# Learning, sleep replay and consolidation of contextual fear memories: A neural network model

Lars Werne[1]*, Angus Chadwick[2], Peggy Seriès[2]*

**1** Center for Doctoral Training: Biomedical AI, University of Edinburgh, Edinburgh, United Kingdom,
**2** Institute for Machine Learning, University of Edinburgh, Edinburgh, United Kingdom

* L.P.J.Werne@inf.ed.ac.uk (LW); pseries@inf.ed.ac.uk (PS)

## Abstract

Contextual fear conditioning is an experimental framework widely used to investigate how aversive experiences affect the valence an animal associates with an environment. While the initial formation of associative context-fear memories is well studied – dependent on plasticity in hippocampus and amygdala – the neural mechanisms underlying their subsequent consolidation remain less understood. Recent evidence suggests that the recall of contextual fear memories shifts from hippocampal-amygdalar to amygdalo-cortical networks as they age. This transition is thought to rely on sleep. In particular, neural replay during hippocampal sharp-wave ripple events seems crucial, though open questions regarding the involved neural interactions remain. Here, we propose a biologically informed neural network model of context-fear learning. It expands the scope of previous models through the addition of a sleep phase. Hippocampal representations of context, formed during wakefulness, are replayed in conjunction with cortical and amygdalar activity patterns to establish long-term fear memories. In addition, valence-coding synapses within the amygdala are subject to homeostatic plasticity overnight, which stabilizes fear associations and regulates the fear circuitry's synaptic density. The model reproduces experimentally observed phenomena, including context-dependent fear renewal and time-dependent increases in fear generalisation. Our model integrates mechanisms of fear learning, systems consolidation and synaptic homeostasis to provide a unified account of how contextual fear memories form and evolve over time. Our framework yields testable predictions about how disruptions in synaptic homeostasis may promote a persistent, fear-sensitized state. Accounting for neural mechanisms that reshape fear memories after their formation is a step towards bridging computational models of fear learning and the mechanisms behind trauma and anxiety disorders.

**Data availability statement:** All code written in support of this publication is publicly available on Zenodo https://doi.org/10.5281/zenodo.15546751.

**Funding:** This work was supported by the United Kingdom Research and Innovation (grant EP/S02431X/1), UKRI Centre for Doctoral Training in Biomedical AI at the University of Edinburgh, School of Informatics. The funders had no role in study design, data collection and analysis, decision to publish, or preparation of the manuscript.

**Competing interests:** The authors have declared that no competing interests exist.

## Author summary

How do we learn to fear certain environments? Why do some fear memories fade while others persist or even grow stronger over time? Scientists have long used laboratory experiments to study how animals associate danger with a particular context. These studies have helped identify brain regions involved in fear learning, including the amygdala, hippocampus, and cortex, and have inspired many computational models of how fear is acquired in the brain. However, most models focus only on what happens when fear is first learned, overlooking how these memories evolve in the following days and nights. In this work, we present a neural network model that captures how fear memories are strengthened or reshaped during sleep. It builds on earlier models by incorporating memory replay and synaptic homeostasis, two brain processes believed to support emotional memory consolidation. Our model identifies neural processes that help make fear memories persistent, suggests that sleep is necessary to maintain adaptive behaviour after threatening experiences, and proposes that sleep disruptions mediate the harmful impact of stress on emotional regulation. By extending amygdala-based models of fear learning to include post-learning processes, we aim to narrow the gap between these models and disorders linked with persistent fear, such as PTSD.

## Introduction

Despite robust biological evidence of sleep's essential role in emotional memory consolidation, computational models of fear learning typically neglect sleep-dependent processes. We propose a neural network model that incorporates sleep-mediated replay and synaptic homeostasis, offering a new mechanistic perspective on fear memory formation, retention, and generalisation.

Pavlovian Fear Conditioning, in which an initially neutral cue or context is repeatedly paired with an aversive, unconditioned stimulus (US), is a fundamental paradigm for investigating the neural mechanisms underlying fear learning [1]. In *contextual* fear conditioning, fear becomes associated with the combined environmental and internal elements defining the conditioned *context* [2]. Understanding how contextual fear memories form, persist, and generalize is crucial both for basic neuroscience and for clinical conditions characterized by maladaptive fear, such as Post-Traumatic Stress Disorder (PTSD) [3,4].

Neural mechanisms underlying fear memory formation extend beyond awake experiences into subsequent periods of sleep. During sleep, neural replay occurs simultaneously across hippocampal and neocortical circuits, reflecting activity patterns learned during prior experiences [5,6]. This coordinated replay has robust experimental support and is critically involved in the consolidation of associative memories, including those formed through fear conditioning [7]. The basolateral amygdala (BLA), long recognized for its central role in emotional learning [8], also

exhibits activity coordinated with hippocampal replay during sleep [9]. Although open questions regarding precise mechanisms and functional significance remain, emerging evidence suggests that coordinated replay across hippocampus and amygdala may help bind contextual representations to emotional salience, driving emotional memory retention [9].

Despite these findings, existing computational models of associative fear learning [10–12], originating from classical frameworks such as the Rescorla-Wagner model [13–15], typically omit these sleep-dependent consolidation mechanisms. These models assume immediate and stable updating of emotional associations during awake learning episodes. As a result, they struggle to account for gradual changes in fear recall over time [16], which may be clinically relevant for the delayed onset or intensification of fear-related symptoms following trauma, as observed in delayed-onset PTSD [17].

To overcome this gap, we propose a new neural network model of contextual fear learning that explicitly incorporates sleep-dependent replay and synaptic homeostasis. Our simulations demonstrate how disruptions in these sleep-related processes – e.g., under psychological stress – could lead to amygdala hyperactivity, enhanced fear acquisition and heightened generalisation. While formulated at a fairly high level of abstraction, our work not only provides a qualitative framework linking sleep to emotional memory processing but also lays a foundation for further computational studies and experimental investigations into fear learning and its clinical implications.

## Methods: Model description

To investigate interactions between sleep processes and fear learning, we developed a computational model that combines a hippocampal sleep replay mechanism, proposed by Fiebig et al. for systems consolidation of episodic memories [18], with an amygdalo-centric framework for associative fear learning, inspired by previous neural network models [11,19]. Building on these proposals, we provide a computational account of how context fear memories transition from transient hippocampal-amygdalar representations into stable amygdalo-cortical associations [20].

Our model reproduces key behavioural phenomena observed in fear conditioning – including context-dependent fear renewal [21] and enhanced fear learning under stress [22,23]. Its architecture is sketched in Fig 1. Its modules represent hypothetical populations of neurons with narrowly defined computational roles. They are named after the brain regions assumed to accommodate neurons with those roles, but the model is not intended to capture the full cellular heterogeneity or connectivity of the implicated regions.

Environmental inputs (or *contexts*) are defined by sampling 50 variables with 10 possible values each. They are represented as binary activity patterns within the sensory cortex (SC) module, from where they are transmitted to each of our model's three primary regions:

- **Hippocampus (HIP)**, which rapidly encodes and temporarily stores context information using Hebbian learning. During sleep, recurrent excitation and fast inhibitory processes drive the replay of these transient engrams.

- **Neocortex (CTX)**, which forms long-term representations through slow synaptic updates. Sleep replay reinforces and stabilizes cortical traces, allowing them to maintain fear associations independently of hippocampal support.

- **Amygdala (BA & CeA)**, which associates contexts with valence. Context identity, fear, and safety are encoded by separate populations of BA cells.

The following sections contain brief descriptions of the computational roles and interactions of the above regions. A more in-depth description of our model architecture is provided in S1 Appendix. In particular, this includes model parameters defining three operational modes – *Perception* (memory formation), *Sleep* (memory consolidation), and *Recall* (memory retrieval), whereas S2 Appendix breaks down the model's update cycle.

**Hippocampal Formation (HIP & EC).** Our implementation of HIP draws on a computational model by Fiebig & Lansner [18]. During *Perception*, environmental inputs activate sparse subsets of HIP cells (4% active), and Hebbian learning strengthens synapses between co-active neurons to form a context-specific *engram*. This plasticity later allows

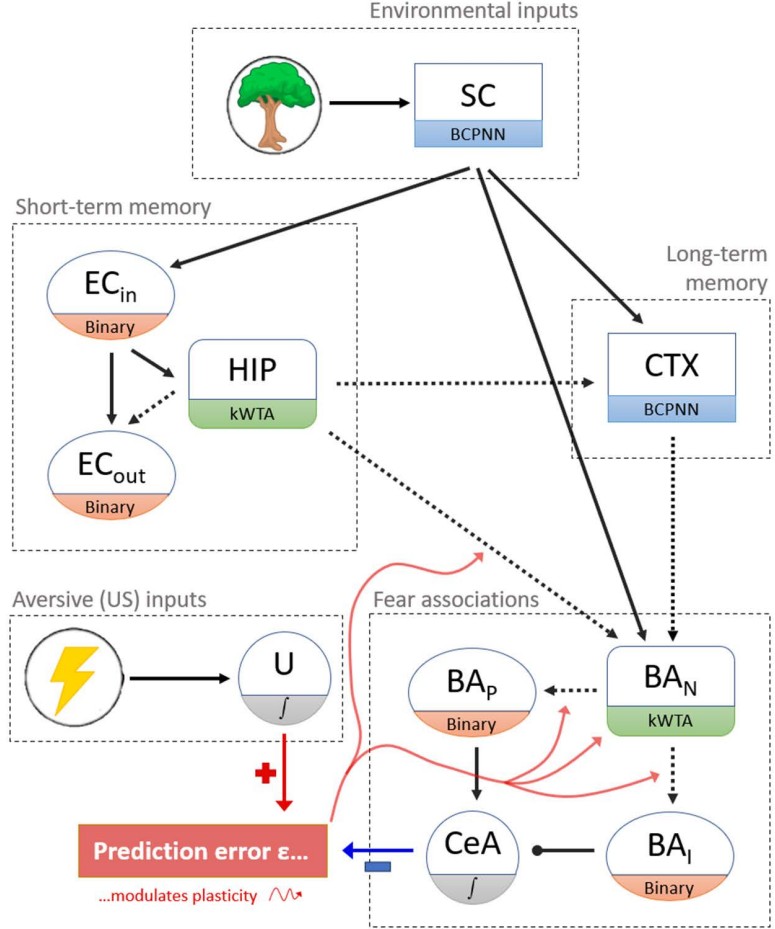

**Fig 1. Model sketch.** Nodes of the graph correspond to *modules*. Dotted/solid arrows denote the existence of plastic/non-plastic synapses between modules. During *Perception*, environmental inputs are passed to the 'engram modules' HIP, CTX and $BA_N$, activating patterns that may be remembered via Hebbian plasticity, so that the model's state may converge onto them during later *Recall*. To allow either fear or safety to be associated with a context, synapses from HIP onto $BA_N$ are strengthened when a prediction error occurs. Prediction errors are defined as the difference between the current US input and fear response (CeA output). Errors promote Hebbian plasticity within $BA_N$ and on HIP→$BA_N$ synapses, allowing the context to be associated with valence. Positive / negative errors further strengthen synapses from $BA_N$ onto $BA_P$ / $BA_I$, increasing / decreasing the amount of fear associated with the current environment. Nodes labelled with a sigmoid are single units. All other modules are sets of neurons – the BCPNN (Bayesian Confidence Propagation Neural Network), kWTA (k-Winner-Takes-All) and 'binary' module types are defined in S1 Appendix. **Abbreviations:** SC = Sensory Cortex. $EC_{(in / out)}$ = Entorhinal Cortex (Input- / Output layers). HIP = Hippocampus. CTX = Neocortex. $BA_{(N / P / I)}$ = Basal Amygdala (valence-**N**eutral / fear-**P**romoting / fear-**I**nhibiting sets of neurons). CeA = Central Nucleus of Amygdala.

HIP to converge onto a relevant engram, when cued during *Recall* (see below). However, hippocampal engrams are relatively short-lived, as they are soon overwritten by new learning [20,24–27]. Because of HIP's sparsity, only the most similar contexts receive overlapping representations [28].

During *Sleep*, in the absence of external inputs, recurrent excitation causes HIP to converge to one of its attractor states – 'replaying' that engram. To prevent the indefinite replay of a single memory, inhibitory synapses between active cells are then rapidly strengthened via a Hebbian learning rule [18]. Eventually, HIP's activity is pushed away, allowing convergence to another attractor. This 'Hebbian depression', or 'adaptation projection' after Fiebig & Lansner [18], has been proposed to abstractly represent a combination of spike frequency adaptation and synaptic short-term depression [24]. Similar to past computational works on neural replay [29–31], HIP in our model thus relies on a combination

of excitatory attractor dynamics and a self-inhibitory fatigue mechanism to cycle through different memory patterns [32]. During *Sleep* replay, plastic connections from HIP to both neocortex (CTX) and basal amygdala ($BA_N$), formed during prior learning, co-activate those regions' corresponding context representations [6].

During *Recall*, HIP activity converges towards the engram of the 'learned' context that is most similar to the current environmental input. Input features of the current environment are compared to those associated with the activated HIP engram via a feedback circuit involving the entorhinal cortex (EC; S1 Fig) [11,33,34]. If input and HIP engram match, the model attempts to retrieve an associated fear response via HIP→$BA_N$ synapses. Otherwise (i.e., if HIP does not recognize the current context), the model instead engages CTX→$BA_N$ synapses, relying on long-term cortical memory to determine if the current environment is associated with fear.

**Neocortex (CTX).** Our model implements the transfer and transformation of context memories from hippocampus to neocortex (*systems consolidation*) for long-term storage [35,36]. Experimental evidence suggests that neocortical engram cells encoding a memory are selected during hippocampal memory formation, but initially lack sufficiently strong connections to be collectively reactivated by sensory cues alone [20]. In our framework, non-plastic synapses from SC to CTX determine the engram cells that go on to encode the current context during *Perception*, while plastic synapses from HIP to CTX are simultaneously strengthened. Therefore, when HIP replays context-specific engrams during *Sleep*, their CTX representations are reactivated and gradually consolidated via Hebbian plasticity. Simultaneously, slowly-evolving synapses from active CTX cells onto co-activated $BA_N$ cells are strengthened. As a result, associations between context and fear responses initially supported by HIP→$BA_N$ synapses progressively shift to stable CTX→$BA_N$ synapses over time (cf. Fig 2; [20]).

Due to its denser activation patterns, CTX engrams overlap more strongly than their HIP counterparts, capturing shared features of remotely experienced environments [37]. Below, we demonstrate that this decrease in memory separation entails increases in fear generalisation over time [16,38–40] (cf. Fig 5d).

**Amygdala (BA & CeA).** Previous neural network models of context fear typically encoded the strength of fear associations directly in synapses connecting hippocampal engram cells to fear-promoting amygdala neurons [10,12,41]. A limitation of this approach is that transferring fear associations from hippocampus to cortex via replay becomes problematic: precise synaptic strengths would need to be copied or accurately reproduced across regions [19]. To address this, our model proposes that the magnitude of fear associations is instead encoded internally within the amygdala, specifically via context-specific amygdalar engram cells that recruit valence-coding neurons through intra-amygdalar

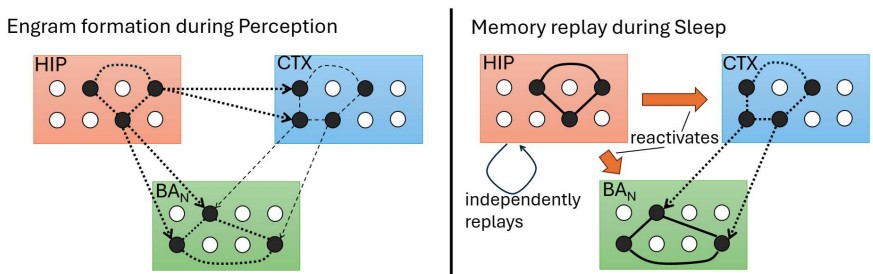

**Fig 2. Dynamics of engram formation and replay in our model.** During *Perception* (left), groups of engram cells in HIP, CTX and $BA_N$ are activated by environmental inputs arriving from SC. Fast Hebbian plasticity (thick, dotted lines) strengthens the HIP and $BA_N$ engrams and forms excitatory connections from the HIP onto the CTX and $BA_N$ ensembles. Plasticity in (and extending from) CTX is slow (thin, dotted lines), not yet supporting later *Recall*. During *Sleep*, HIP independently replays stored engrams, co-activating its CTX and $BA_N$ counterparts. This allows the CTX engram to consolidate and to form connections onto the associated $BA_N$ engram. In this way, sleep replay drives the long-term retention of fear (and safety) memories in our model.

synapses [20,42]. Thus, coordinated sleep replay only needs to associate cortical context engrams with corresponding amygdala engrams – a simpler task (cf. Fig 2).

In our model, when a *prediction error* occurs (i.e., a difference between the current aversive input (US) and the model's fear response), recurrent synapses between active valence-neutral amygdala engram cells ($BA_N$) are strengthened [43,44], providing $BA_N$ with a stable index of the current context. Depending on its sign, the error signal further drives a strengthening of synapses extending from this $BA_N$ engram onto separate populations of fear-promoting ($BA_P$) or fear-inhibiting ($BA_I$) cells [45,46]. In this way, valence-coding neurons can persistently become recruited into a context representation. Active $BA_P$ (or $BA_I$) neurons drive the model's fear response by exciting (or inhibiting) its single fear-output neuron, analogous to the central nucleus of the amygdala ('CeA') [47,48].

### Recruitability

$BA_P$ cells that have become associated with many different contexts may, as a result, receive strong excitation even in entirely novel environments – leading to indiscriminate fear expression. To prevent this scenario, our model limits synaptic plasticity during each learning episode to a small, time-varying subset of valence-coding neurons. This modulation is controlled by a cell-specific factor termed 'recruitability' ($\mathcal{R}$). During fear or safety learning, the prediction error gating plasticity onto each valence-coding neuron (in $BA_P$ or $BA_I$) is scaled by its current $\mathcal{R}$ value. At any given time, $\mathcal{R}$ is close to 1 for only a few cells, which are then more likely to undergo plasticity (Fig 3a).

Our implementation of this mechanism (S2 Fig; cf. S2 Appendix) was computationally motivated, and it does not reflect a specific biological signal. Nonetheless, the brain likely employs various processes to regulate neuronal allocation [49]. For example, neurons expressing higher levels of the transcription factor CREB exhibit increased excitability and preferentially become assigned to newly formed memory traces [50–52]. Our implementation reflects the finding that fear memories formed close in time to each other show greater neural overlap [51,52]. Although this finding also holds outside the amygdala – and for memories not related to fear [52,53] – we only apply the above recruitability mechanism to the valence-coding $BA_P$ and $BA_I$ regions. For simplicity, which cells form context-coding engrams in *HIP*, *CTX* and $BA_N$ depends entirely on the features of the context presented during memory formation.

**Homeostasis.** The *synaptic homeostasis hypothesis* proposes that sleep globally weakens synapses to preserve functionally meaningful connections [54]. Experimental evidence confirms that synaptic homeostasis during sleep is selective, preferentially weakening synapses that are already weak, while relatively preserving stronger ones, as observed in sensory and motor cortices [55,56]. Within the amygdala, sleep-dependent synaptic adjustments appear spine-type- and subregion-specific, though their precise functional implications remain unclear [57]. Nevertheless, these observations are intriguing given that sleep deprivation increases amygdala reactivity [58] and may facilitate future fear acquisition [59]. These findings raise the possibility that disruptions of synaptic sleep homeostasis promote amygdala hyperexcitability, contributing to maladaptive fear sensitization and generalisation – phenomena central to pathological anxiety [60].

In our model, synaptic homeostasis at connections onto valence-coding amygdala cells ensures adaptive fear learning. For valence to be stably associated with a context, synapses between context engram cells ($BA_N$) and valence-coding cells (P- or I-cells) typically require repeated strengthening over multiple subsequent simulation steps during *Perception*. Although slow fluctuations in cell-specific recruitability promote the recruitment of specific neurons in specific conditioning events, random noise occasionally produces transiently high recruitability in some cells for a single time step (Fig 3a). This can produce partially strengthened, functionally ineffective synapses that do not reflect stable context-valence associations. Unregulated, such incidental synaptic changes could accumulate, resulting in inappropriate fear responses or accelerated acquisition (interpreted as non-associative fear sensitization [22]). Furthermore, excessively strong synapses could lead to inappropriate fear generalisation, as will become clear in later results.

To mitigate these issues, we implemented a synaptic homeostasis mechanism modelled via a *cubic growth function*. Although cubic growth has not previously been proposed as a biological model of synaptic change, our implementation

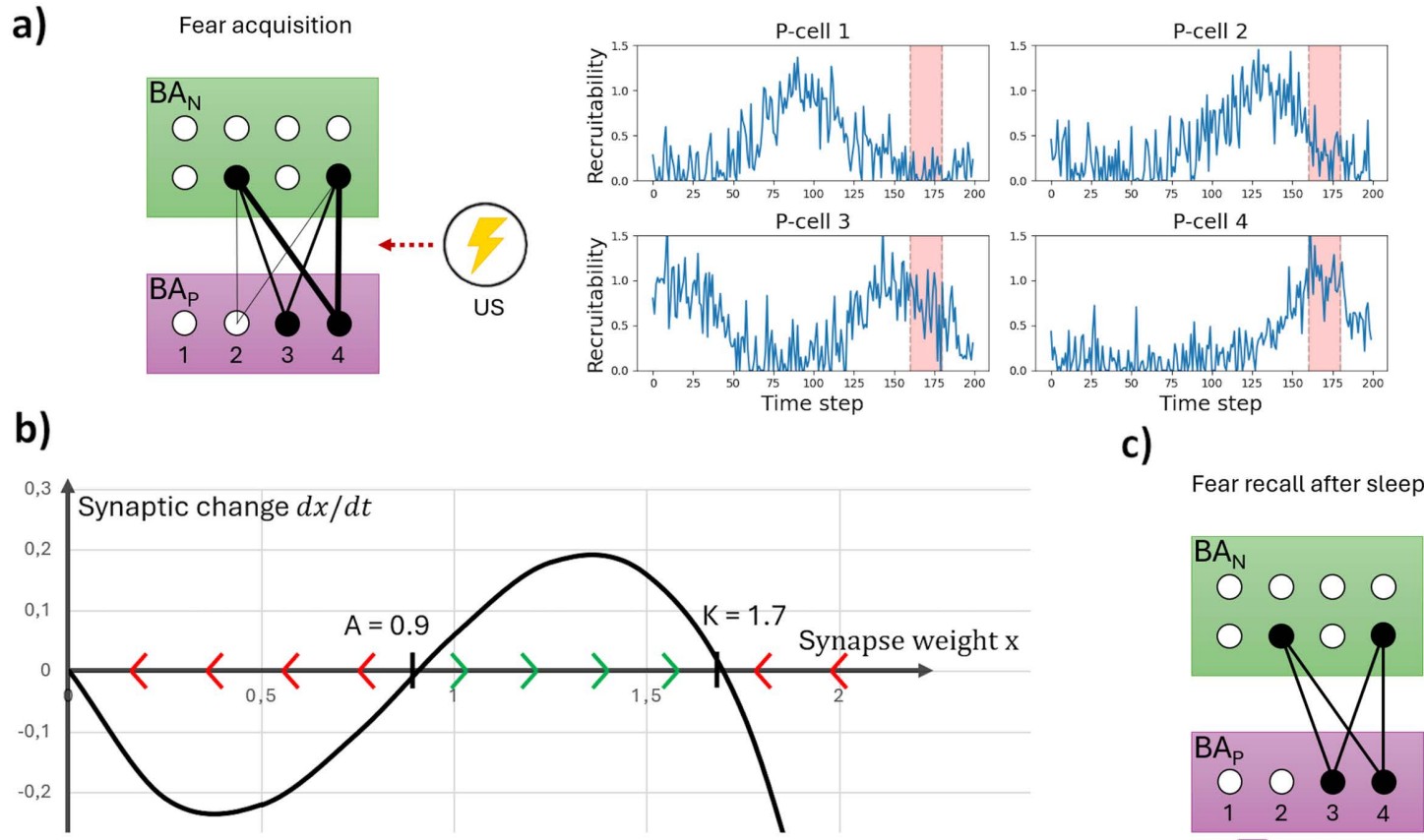

**Fig 3. Fear acquisition and synaptic homeostasis. a) Fear acquisition:** When a surprising US occurs, active (valence-Neutral) $BA_N$ engram cells, encoding the current context, become associated with fear. **Left:** Over the course of several time steps, synapses from the $BA_N$ engram onto fear-Promoting $BA_P$ cells are strengthened by various amounts (denoted by the different line weights). **Right:** The extent to which synapses onto certain P-cells are strengthened is mediated by those cells' current *recruitability* (S2 Fig). In this example, the time window of US delivery (time step $160 - 180$) is shaded in red. P-cells 3 and (especially) 4, but not 1, are recruited – i.e., become associated with the current context – as the US signals coincide with their time windows of high recruitability. Synapses onto P-cell 2 are strengthened to some, limited extent as random noise has raised its recruitability for a single time step during conditioning. **b) Cubic growth model:** The differential equation describing homeostatic changes applied to the strength of our model's valence-coding synapses during *Sleep*, $\frac{dx}{dt}(x) = -rx(1 - \frac{x}{K})(\frac{x}{A} - 1)$ with (exaggerated) learning rate $r = 1.4$, extinction threshold $A = 0.9$ and recruitment strength $K = 1.7$. For $t \to \infty$, the strength stably converges to $x = 0$ or $x = K$, depending on whether its starting value $x(0)$ lies above or below $A$. This rule gives rise to the synaptic changes from **a)** (Left) to **c)**. **c) Fear recall after sleep:** When the conditioned context is encountered after a *Sleep* phase, synaptic homeostasis has acted on the synapses strengthened in **a)**. Weak synapses have been pruned, whereas stronger synapses have been normalized towards an appropriate strength. See also Fig 7 and its discussion in later results.

reflects general experimental observations: during sleep, weak or incidentally strengthened synapses are preferentially pruned, while strong synapses are relatively preserved [55,56] and may even be strengthened [61,62] (Fig 3c).

In our model, synaptic sleep homeostasis is only applied to synapses onto the valence-coding $BA_P$ and $BA_I$ modules, where it affects fear expression. In reality, biochemical mechanisms protecting specific synapses from degradation may play key roles more broadly, for instance for the retention of long-term memories [63].

## Results

In the following, we demonstrate our model's behaviour across a range of simulations that target different facets of fear memory formation, consolidation, and retrieval. All simulation protocols are provided in S3 Appendix. A full list of the

model's default parameter values is given in Table C of S1 Appendix. Unless stated otherwise, these values are consistent across all reported simulations.

## Context engram formation, replay and recall

In *Perception* mode, the activity patterns of our model's engram modules HIP, CTX and $BA_N$ depend on the features of the current environmental input. All three modules are capable of storing their activity patterns by strengthening excitatory synapses between co-active units, which thereafter form an engram that may be re-activated in *Recall* mode. Fig 4a illustrates the dynamics of engram formation in a context presented for 50 time steps. HIP swiftly formed a strongly connected engram, whereas CTX advanced much more slowly. $BA_N$ was able to form an engram very rapidly, but only did so once a surprising US was delivered. At the same time, the model formed synapses from active HIP- to active CTX- and $BA_N$ units, which would lay the foundation for coordinated replay across all three regions in a subsequent *Sleep* phase.

Such sleep replay is illustrated in Fig 4b. This plot demonstrates the hippocampal replay of 10 contexts, previously shown to the model, over the course of a *Sleep* phase. Whenever HIP replayed a context, the corresponding CTX pattern was co-activated. $BA_N$ participated in this coordinated replay only for contexts for which it had formed an engram – i.e., those which had been paired with US signals (contexts 6–10). Qualitatively, this aligns with the observed participation of the amygdala during the hippocampal replay of recent, threatening experiences in rats [9].

In our simulations, the default duration of *Sleep* is 165 time steps. As a functional abstraction, we leave the duration of an individual time step indeterminate. In reality, hippocampal replay is primarily observed during sharp-wave ripple events, which last tens to hundreds of milliseconds and occur dispersed throughout sleep [64,65]. In the following simulations, *days* refer to time windows the model spends in *Perception* and/or *Recall* mode (one or several experimental sessions). Days are interleaved with blocks of *Sleep*.

During *Sleep* replay, Hebbian plasticity drives the strengthening of synapses between the CTX and $BA_N$ engrams of emotionally salient contexts. This allows the $BA_N$ ensemble to be recalled even after the corresponding HIP engram is forgotten. The recall performance of our three engram modules is demonstrated in Fig 4c. In this simulation, the model perceived three novel contexts on each of 25 days, intermitted by *Sleep* phases. Each context was joined by light US signals to engage plasticity in $BA_N$. To assess the model's ability to remember each context, it was ultimately placed in *Recall* mode. Each of the 75 original input patterns was 'blurred' (10% random masking) and presented to the model for one time step. If, after 30 time steps without further inputs, a module's activity matched the original memory trace (within some margin of error), recall was 'successful'. HIP consistently recalled recent contexts but failed to retrieve context memories older than approximately 10 days. In contrast, CTX's ability to recall contexts initially increased over time, rising steeply after one night of sleep and peaking three to five days post-encoding. The initial rise reflects ongoing consolidation of CTX engrams driven by hippocampal sleep replay. $BA_N$, receiving inputs from both HIP and CTX, successfully recalled contexts as long as at least one of these regions had retained the associated memory.

In summary, replay of recently formed HIP memory traces during *Sleep* co-activates associated CTX and $BA_N$ engrams. Hebbian plasticity strengthens the CTX engram's synapses and links it to the co-activated $BA_N$ trace. After one or several such replay events, this mechanism allows the *Recall* of that context's $BA_N$ engram, when cued with a relevant input, even if the initial HIP encoding has grown too weak to be retrieved.

## Fear acquisition and extinction

For our model to explicitly associate a context with fear (or safety), it must form a $BA_N$ engram for that context, attach it to the corresponding HIP trace, and learn excitatory synapses onto fear-promoting $BA_P$ (or fear-inhibiting $BA_I$) cells. Fig 4d shows a simple simulation of fear acquisition and extinction in the same environment. When a context was repeatedly paired with a US, the model rapidly developed a fear response by recruiting $BA_P$ cells. Once the aversive stimulus was

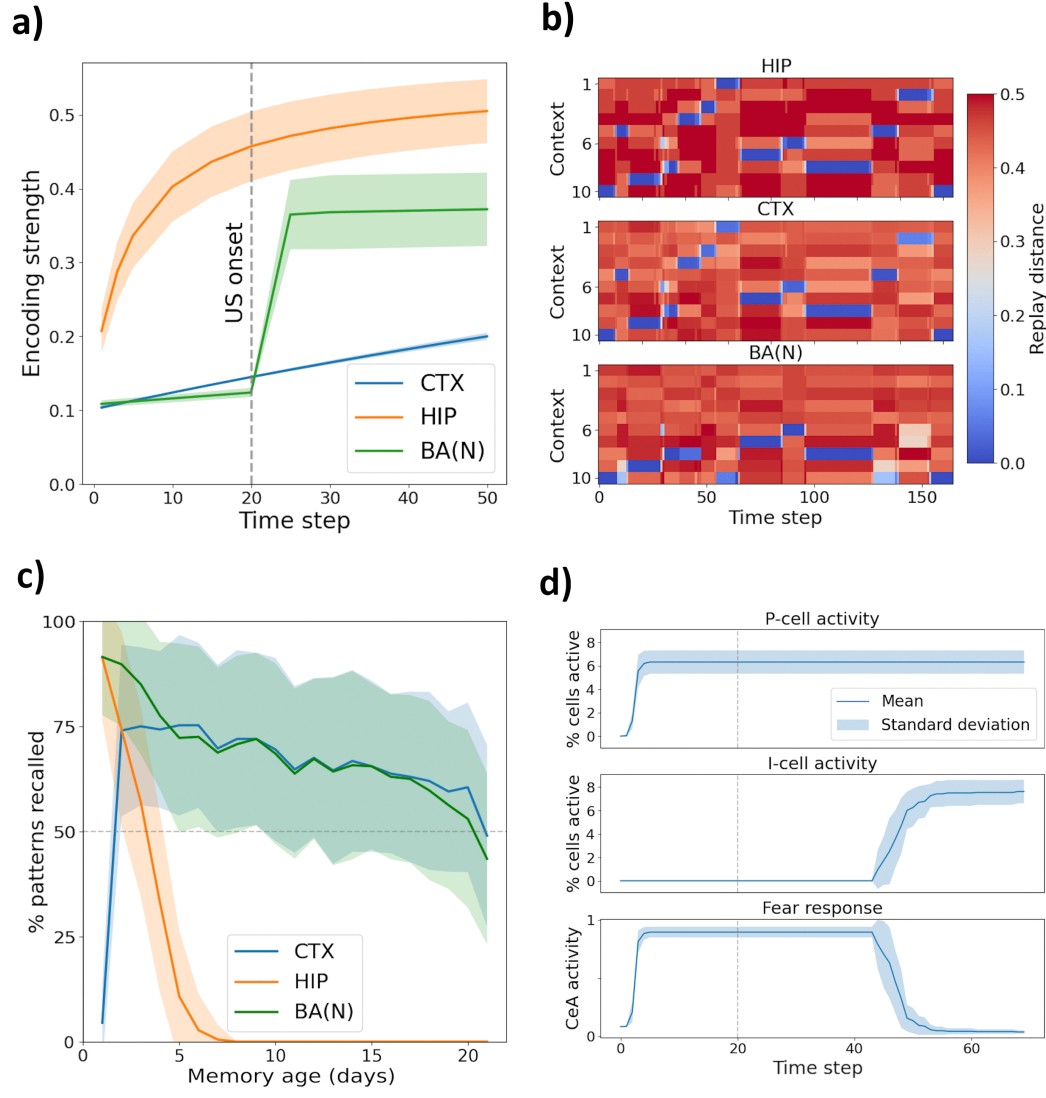

**Fig 4. Mechanisms of context/fear memory formation. a) Engram formation:** Encoding strength (net synaptic weight between active engram cells, normalized by their total outgoing weights) of a novel context, presented for 50 time steps, in HIP, CTX, and BA_N. US inputs occurred on the last 30 time steps. HIP rapidly formed context engrams; BA_N did so after the first US; CTX learned slowly. Results were averaged over 10 simulation runs. Shaded regions denote $\pm 1$ standard deviation (SD) across runs. **b) Sleep replay:** Distance between original context patterns and neural activity during *Sleep* in HIP, CTX, and BA_N following sequential exposure to 10 novel contexts. In contexts 6 to 10, US signals were delivered. Blue indicates replay of stored engrams. All contexts were replayed in HIP and CTX; BA_N replay was limited to contexts paired with aversive stimuli. The distance measure was borrowed from Greve et al. [68]. **c) Recall performance:** Percentage of contexts successfully recalled by HIP, CTX, and BA_N as a function of days since initial encoding. HIP reliably retrieved recently, but not remotely, perceived contexts. CTX was most likely to recall contexts encoded several days ago, reflecting sleep-dependent consolidation. Compared to HIP, CTX featured a recall curve with a lower peak, but flatter drop-off, indicating a more selective, more durable, storage strategy. BA_N recalled contexts when supported by either HIP or CTX. Curves were averaged over 100 runs, with 4 contexts assessed per run and day. Shaded regions denote $\pm 1$ SD on the percentage of recalled contexts across runs. **d) Acquisition and extinction:** Fear acquisition and extinction dynamics. In each of 30 simulation runs, a context was presented for 70 steps, with US inputs delivered only during the first 20. The model acquired fear quickly (via BA_P activation), which was slowly extinguished following removal of the US through gradual activation of BA_I cells. BA_N→BA_I synapses began growing in strength as soon as the US signals were removed, but fear did not decrease until – after a delay – the first BA_I cells became active. Shaded regions denote $\pm 1$ SD across runs.

removed, continued exposure to the conditioned environment gradually strengthened $BA_N \rightarrow BA_I$ synapses until, after a delay, this caused fear to be extinguished.

Our choice of modelling fear extinction through the recruitment of fear-inhibiting neurons, downstream of the encoding for the fear association, was informed by neurobiological findings [45,46] and the observation that – at least during early stages of extinction [66,67] – the brain primarily silences learned fear associations but initially does not erase them. Phenomenologically, this is also evidenced by the spontaneous return of previously extinguished fear, as well as context-dependent fear renewal effects – which our model reproduces, as outlined in the following section.

## Fear renewal and generalisation

In our model, fear (or safety) associations can generalize because contexts with shared input features receive overlapping engrams in HIP, CTX and – in particular – $BA_N$. Hence, valence-coding cells linked with a conditioned context may become active when a similar environment is presented in *Perception* or *Recall* mode – if the input they receive from $BA_N$ cells shared between the contexts is sufficient (cf. Fig 5a). In reality, fear extinguished in one environment often returns when encountering a novel, but similar context. This context-dependent return of fear is called *fear renewal*. According to a common interpretation, this occurs because extinction memories are more *context-specific* than fear memories [69].

We captured this asymmetry by assigning higher activation thresholds to fear-inhibiting (I-) cells compared to fear-promoting (P-) cells. Our model's Hebbian learning rule places an upper bound on the strength of any synapse [24]. This allowed us to tune the model such that $BA_I$ neurons, when they are recruited, remain just slightly above their activation threshold. Hence, after moving to a similar but different context, they are unlikely to stay active. In contrast, during fear learning, plasticity on some $BA_N \rightarrow BA_P$ synapses may continue after the postsynaptic cell has become active, as long as a prediction error persists. The excess excitation received by those $BA_P$ cells later causes them to activate even in moderately similar contexts.

To illustrate this, we trained the model on fear acquisition followed by extinction in a specific context ('A'). Subsequently, we assessed the activation of P- and I-cells when the model encountered contexts with varying degrees of similarity to 'A' (Fig 5b). Contexts with $x\%$ similarity to 'A' were generated by fixing a random $x\%$ of its 50 input features and independently re-sampling the remainder. As intended, I-cells recruited during extinction required contexts to be highly similar (above 80% input overlap) to become reactivated. In contrast, P-cells could reactivate in contexts with as little as 50% input overlap with 'A', causing fear responses to generalize broadly. These dynamics led to realistic patterns of fear renewal across classical ABA, ABC, and AAB protocols, in which fear is first acquired (in context 'A'), extinguished (in 'A' or 'B'), and tested for renewal (in 'A', 'B', or a novel context 'C'; Fig 5c). Notably, renewal magnitude followed the order $ABA > ABC > AAB$, in line with experimental observations [21,70]. In all cases, extinction in renewal contexts occurred faster than initial extinction, reflecting sub-threshold excitation accumulated onto I-cells during prior extinction learning [71,72].

Our model also captures increases in fear generalisation over time, consistent with experimental reports [16,38–40]. This emerges naturally through the transition from HIP- to CTX-dependent recall. Immediately after fear acquisition in context 'A', exposure to a moderately similar but harmless context 'B' initially does not elicit significant fear. After a single *Sleep* phase, fear expression in 'B' even ceases entirely (cf. Fig 7). However, as the context memories consolidate over time and retrieval increasingly relies on cortical representations - less sparse and more overlapping than their hippocampal counterparts [18], fear expressed upon *Recall* of context 'B' gradually increases (Fig 5d). This occurs because the cortical engram for context 'A' becomes partially activated when remotely recalling context 'B', thereby activating portions of the conditioned $BA_N$ engram. Importantly, because $BA_N$ forms specific engrams only for contexts linked with prediction errors during *Perception*, systems consolidation selectively increases fear expression in context 'B' without diminishing fear recall in 'A'. Our model thus aligns with the experimental observation that fear generalisation increases over time due to decreased memory specificity upstream of the amygdala – yet crucially, the original fear memory itself remains robustly recallable, consistent with empirical findings [40,73,74].

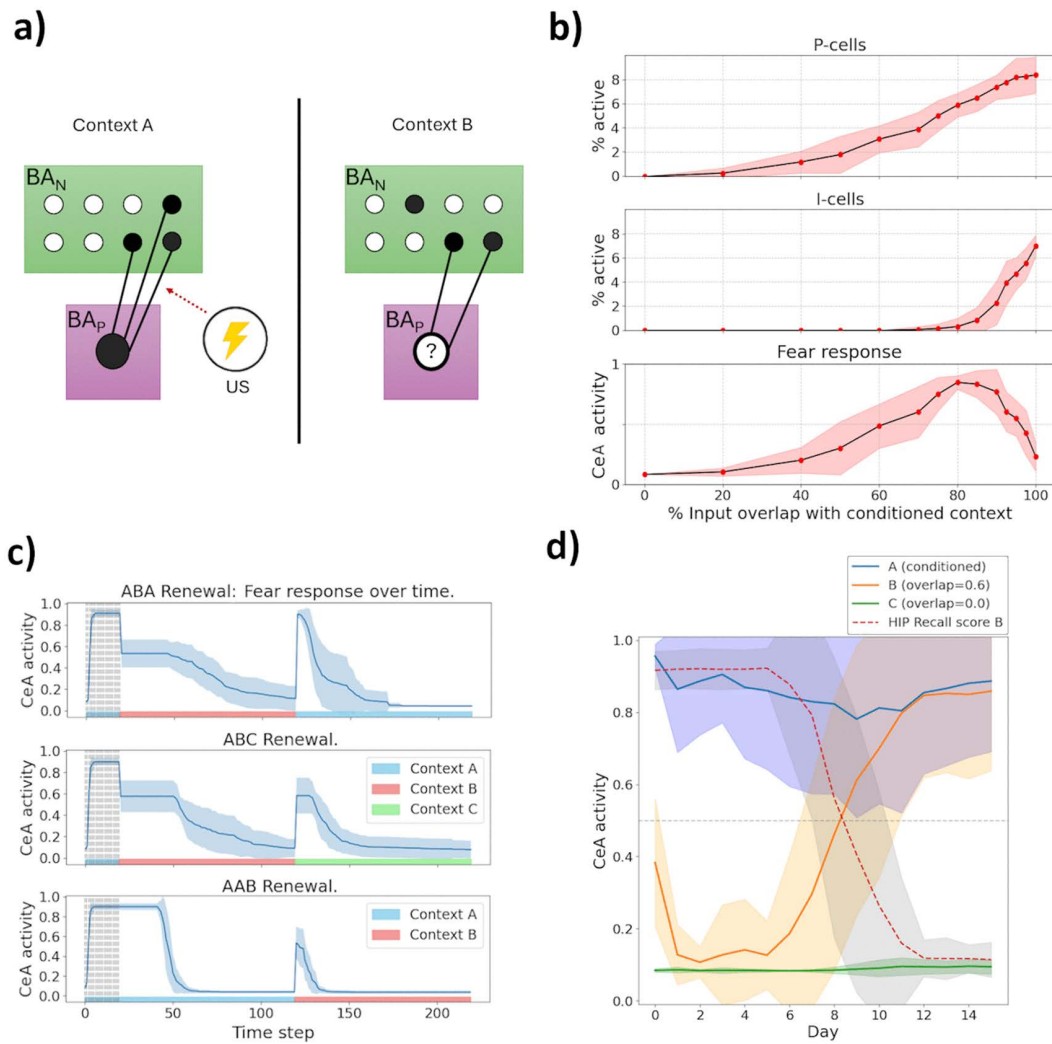

**Fig 5. Results on fear generalisation. a) Generalisation schema.** Fear conditioning in context 'A' strengthens synapses between its $BA_N$ engram and $BA_P$ cells, activating the latter. In a similar context 'B', some of these $BA_N$ cells are active, exciting the same $BA_P$ cells. Whether these become active, causing fear to generalize, depends on the margin by which their activation threshold was exceeded in context 'A' and on the overlap of the two $BA_N$ engrams. **b) Generalisation gradients.** After fear acquisition and extinction in context 'A', red dots denotes the fraction of active $BA_P$-cells (top), of active $BA_I$-cells (center) or the magnitude of the CeA output (bottom) when the model meets a context whose feature overlap with 'A' is shown on the x-axis. Plots were averaged over 10 runs. Fear renewal is likeliest to occur when the model is placed in a context moderately similar to 'A'. **c) Renewal paradigms.** From top to bottom, the figure shows the activity of our model's CeA cell when subjected to the ABA, ABC and AAB fear renewal paradigms, as described in the main text. Plots were averaged over 10 runs. **d) Fear generalisation increases over time.** After fear conditioning in context 'A', fear expressed in a similar, unconditioned context 'B' increases as days pass. In a dissimilar context 'C', no fear is expressed, showing that the increase in 'B' is associative. The dotted 'Recall score' (F1 measure) quantifies how accurately the HIP engram retrieved when observing context 'B' in *Recall* encodes its true input features (S1 Fig) – it is inversely proportional to fear expression in 'B', indicating the generalisation increase is tied to use of the CTX→$BA_N$ pathway for *Recall*. Curves were averaged over 50 runs. In panels b, c and d, shaded regions denote $\pm 1$ SD across runs.

## Fear extinction in multiple contexts

Fear extinction is thought to be the primary mechanism underlying exposure therapy for anxiety disorders [75]. Correspondingly, fear renewal poses a major challenge to clinicians [76], and potential strategies for preventing it are of great clinical interest [77]. Out of the protocols introduced in the previous section (Fig 5c), *ABC* renewal is particularly relevant

to exposure therapy. That is because, generally, neither fearful reminders in the clinic, nor those that may later spark fear renewal, quite match the context in which fear was originally acquired [78].

Here we asked whether distributing extinction across multiple contexts $B_1, \ldots, B_k$ is helpful for suppressing $ABC$ renewal in our model – a strategy that has previously been proposed [78–80]. To this end, we compared two protocols matched for total duration of extinction training following acquisition in context $A$:

1. single–context extinction in $B$ (with 80% input overlap to $A$); and

2. multi–context extinction in four independently sampled contexts $B_1, \ldots, B_4$ (each with 80% input overlap to $A$).

Fig 6a shows the model's average CeA output across the three phases; in the multi–context condition, fear at the onset of testing in $C$ was lower than in the single–context condition. In other words, fear extinction generalized better to the novel context when it was repeated in several contexts to which fear had generalized.

Here, this effect occurred because, when our model entered the second, third or fourth extinction context, there generally was some return of fear. These spikes in prediction error, which did not occur in the single-context protocol, caused an additional strengthening of $BA_N \rightarrow BA_I$ synapses, for those $BA_N$ cells shared between different extinction environments. In turn, stronger $BA_N \rightarrow BA_I$ synapses meant that $BA_I$ cells were more likely to activate in the novel context $C$.

Next, we varied the $A \leftrightarrow B_i$ similarity while holding $A \leftrightarrow C$ fixed at 80%. An *intermediate* similarity ($A \leftrightarrow B_i \simeq 0.80$–$0.85$) minimised renewal in $C$ (Fig 6b; S3 Fig). Intuitively, if the $B_i$ were too similar to $A$ ($A \leftrightarrow B_i \geq 0.9$), it was as if extinction had been performed exclusively in $A$ – allowing fear renewal, as in the $AAB$ protocol. If $A$ and the $B_i$ were too dissimilar, the $BA_N$ engrams of the extinction contexts did not share a sufficient overlap – amongst each other, nor with $C$ – to cause a compounding effect that would have allowed extinction to generalize.

Fig 6b further shows the impact of varying $A \leftrightarrow B_i$ for $A \leftrightarrow C = 0.6, 0.7, 0.9, 0.95$. Interestingly, when acquisition and renewal contexts were more similar, fear renewal was easier to prevent, and larger values of $A \leftrightarrow B_i$ became ideal for achieving this. On the other hand, for $A \leftrightarrow C = 0.6$ or $0.7$, there was only moderate fear expression in context $C$ – which, however, was impossible to abolish entirely using the present protocol. No matter which $A \leftrightarrow B_i$ overlap was chosen, extinction did not generalize to $C$.

The above results should not be taken as precise quantitative predictions, since they depend on parameter choices controlling, e.g., $BA_{PII}$ generalisation gradients, as well as on our definition of 'context similarity' (cf. Methods: Model description). Nonetheless, an interesting suggestion is that the similarity among acquisition, extinction and renewal contexts can strongly affect the efficacy of distributing fear extinction across multiple contexts. To illustrate the intuition behind this prediction, independently of implementation details, we reproduce the qualitative shape of the curves in Fig 6b (for $A \leftrightarrow C = 0.8, 0.9, 0.95$) using a toy model, in S1 Text. These considerations may be important for reconciling mixed results on the benefit of multi-context extinction [81,82].

## Sleep homeostasis

Besides memory consolidation, sleep in our model plays a critical role in maintaining synaptic homeostasis within the fear circuitry. In any fear conditioning event (say, in context 'A'), synapses from active $BA_N$ cells onto fear-promoting (P-) cells are strengthened to varying degrees, modulated by each P-cell's current recruitability (Fig 3a, S2 Fig). If the model later encounters a novel, similar context 'B', any P-cell receives a fraction of the excitation it did in context 'A' (Fig 7a,b). Even if the most strongly innervated P-cells initially remained below activation threshold, their subthreshold excitation would bias the model towards recruiting the same cells again to rapidly acquire fear in context 'B', should a mild aversive event occur. If this consistently occurred in several environments, this cascading bias would soon cause the activity of the affected P-cells to generalize to a very broad range of contexts.

While enhanced fear learning after stressful experiences is plausible [22,23], unchecked, the above effect may eventually cause indiscriminate fear expression. Addressing this is one role of our synaptic homeostasis mechanism (cf.

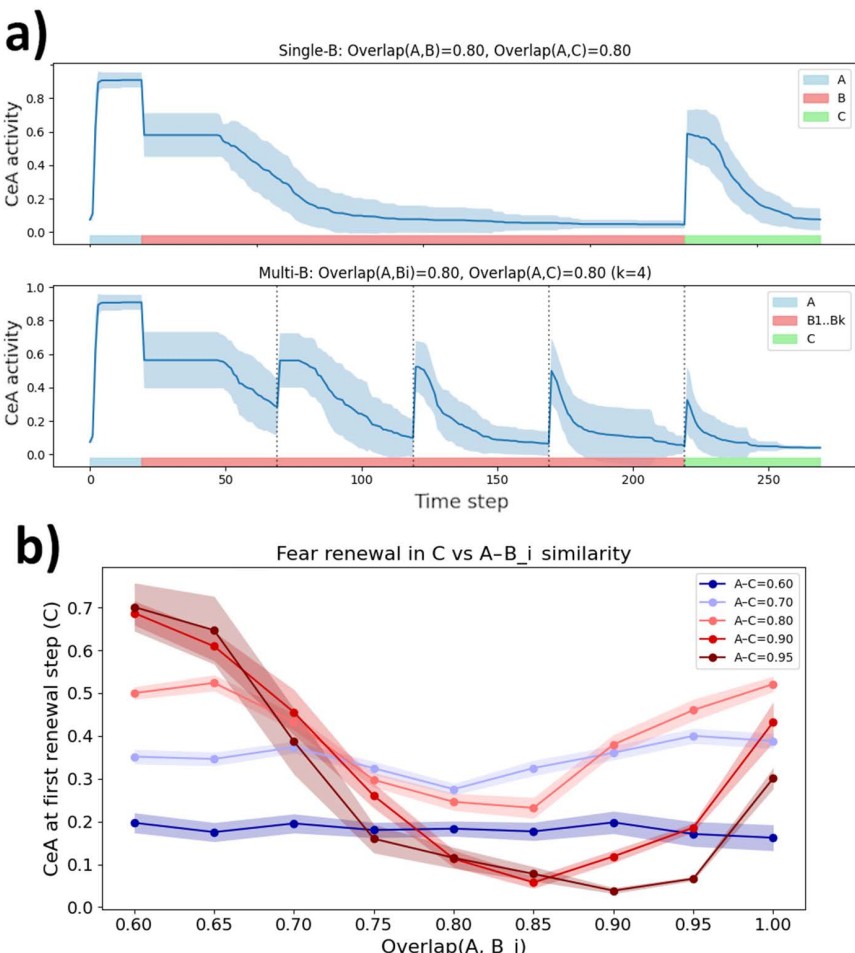

**Fig 6. ABC Renewal with extinction in multiple contexts. a) Multi-context extinction suppresses renewal of conditioned fear.** Each panel shows the average CeA output across acquisition (context *A*, blue), extinction (*B*, red), and renewal (*C*, green) phases. **Top:** Single-context extinction protocol ($A{\to}B{\to}C$) with 80% input overlap between *A* and *B*, as well as between *A* and *C*. **Bottom:** Multi-context extinction ($A{\to}B_1{\ldots}B_4{\to}C$), where each $B_i$, as well as *C*, shares 80% overlap with *A*. Shaded regions show ±SD across runs ($n=10$). In the multi-context condition, fear expression (CeA activity) at the onset of testing in *C* is significantly lower than after single-context extinction, suggesting that distributing extinction across several similar but non-identical contexts can reduce fear renewal. **b) Fear renewal in *C* as a function of $A{\leftrightarrow}B$ similarity.** We performed the multi-context renewal paradigm from subplot a), varying the input overlaps $A{\leftrightarrow}B_i(i=1,2,3,4)$ and $A{\leftrightarrow}C$. Different lines correspond to different values of $A{\leftrightarrow}C$; points represent means across ($n=50$) runs; shaded bands denote ±SEM. The rightmost points ($A{\leftrightarrow}B_i=1.0$) correspond to extinction in *A*. Generally, distributing extinction across multiple contexts lowers fear renewal in *C*. For renewal contexts very similar to *A*, using a set of $B_i$ that share a large overlap with *A* is most effective. However, as the similarity between *A* and *C* decreases, introducing greater differences between *A* and the $B_i$ becomes beneficial, since additional variation in the $B_i$ allows extinction to generalize more broadly. In the lowest similarity condition ($A{\leftrightarrow}C=0.6$), renewal is moderate, but unaffected by extinction training, as the $B_i$ are too dissimilar from *C* for extinction to generalize.

'Methods: Model description') – it depresses excessively strong synapses onto P-cells, thereby lowering the tendency of their recruitment to generalize. As illustrated in Fig 7c-d, this homeostatic adjustment reduces the net innervation of 'recruited' P-cells in context 'A' towards a stable intermediate range, limiting associatively generalized fear in 'B'. Additionally, the same homeostatic process also prunes weak synapses onto P-cells, to prevent a non-associative buildup of excitatory inputs over time – an effect we further examine in the following.

**Sleep deprivation.** Insomnia – a chronic difficulty in falling or staying asleep – is common across anxiety disorders [84,85]. In the aftermath of traumatic events, sleep disruption is a strong predictor of later PTSD, though causal links have

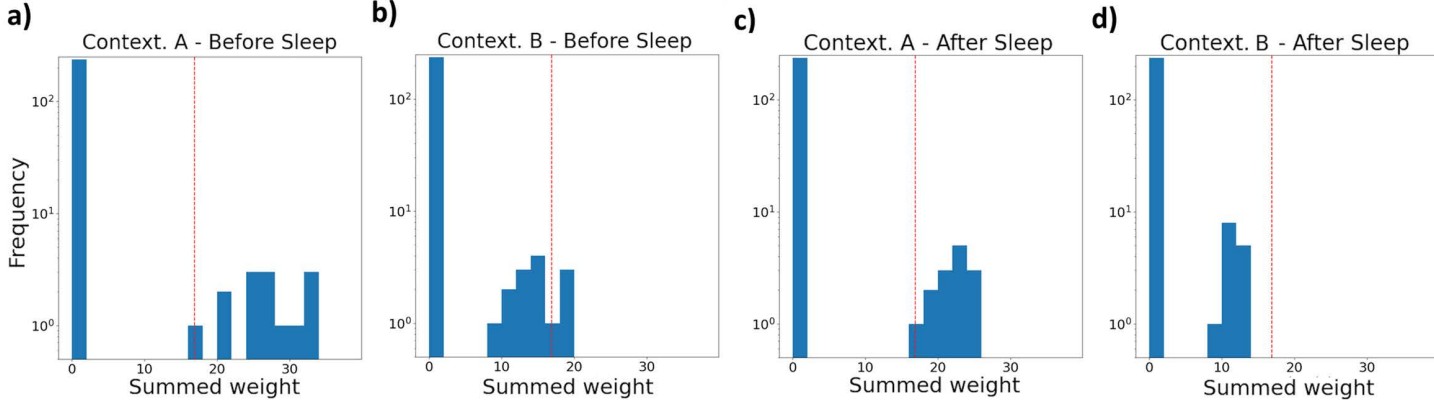

**Fig 7. Histograms of the summed strength of afferent synapses from active BA$_N$ units over all P-cells.** The red lines symbolize the least amount of excitation a P-cell must receive to become active. Histograms were computed: **a)** at the end of fear acquisition in context 'A'. BA$_P$ cells falling on the right of the red line are active, sparking fear. **b)** upon being immediately placed in an unconditioned, vaguely similar context 'B'. The BA$_P$ cells that were most strongly innervated in 'A' are active – expressing some generalized fear. **c)** upon revisiting context 'A' after a *Sleep* phase. The amount of fear expressed has not changed relative to a) but – thanks to homeostatic synaptic adjustments during *Sleep* – no BA$_P$ cell receives much more excitation than needed to be activated. **d)** upon revisiting context 'B' after a *Sleep* phase. Fear is no longer expressed in this unconditioned context. A selective normalization / partial reversal of recently strengthened fear-coding synapses in amygdala circuits during *Sleep* is thus posited to underlie decreases in the generalisation of freshly learned fear associations [83].

remained unclear [86,87]. Fear conditioning studies have pointed at ways in which sleep omission may exacerbate fear memories. For instance, sleep between sessions has been shown to counteract sensitization to experimental stressors [86,88,89], and sleep deprivation may enhance fear expression in both rats [90] and humans [83,91]. In contrast, other studies have proposed deliberately *avoiding* sleep in the aftermath of trauma, as a means of interfering with memory consolidation and preventing later PTSD [92,93]. Clarifying the involved neural mechanisms will be key for reconciling these lines of evidence.

In our model, sleep offers protection from fear sensitization – exaggerated fear learning and expression – by driving a regular pruning of weak $BA_N \rightarrow BA_P$ synapses. This process limits the accumulation of redundant synapses, which may form as a byproduct of fear conditioning or in response to negligible US signals. Our framework thus suggests a role of sleep in moderating the synaptic density of the amygdala fear circuit and, with it, an individual's tendency towards fear acquisition and expression.

To demonstrate this prediction, we designed a simulation that mimics daily experiences alongside sleep-induced synaptic homeostasis. Over seven days, the model encountered three novel contexts each day, paired with moderate US signals. Between days, the model underwent a *Sleep* phase. On the eighth day, the model was briefly exposed to a novel context, receiving a US (of strength 0.75) for three consecutive time steps, after which its fear response was recorded. Repeating this protocol while systematically reducing the duration of the *Sleep* phases revealed that shortened sleep resulted in *fear sensitization*. Fear acquisition in the novel context was accelerated (Fig 8a), accompanied by net increases in $BA_N \rightarrow BA_P$ synaptic weights (Fig 8b).

To illustrate that *Sleep*'s impact on future fear acquisition is mediated by our model's homeostasis mechanism, we repeated the above procedure for several variants of our model, with altered parameters (cf. S2 Text). When we increased (decreased) the rate of homeostatic plasticity on $BA_N \rightarrow BA_P$ synapses, fear acquisition was weakened (strengthened) for all *Sleep* durations, without disrupting the monotonic relationship between *Sleep* length and fear sensitization. On the other hand, when we lowered the minimum strength a $BA_N \rightarrow BA_P$ synapse must have to be consolidated rather than pruned by the homeostasis mechanism (the *extinction threshold*, cf. Fig 3b), extended *Sleep* durations enhanced

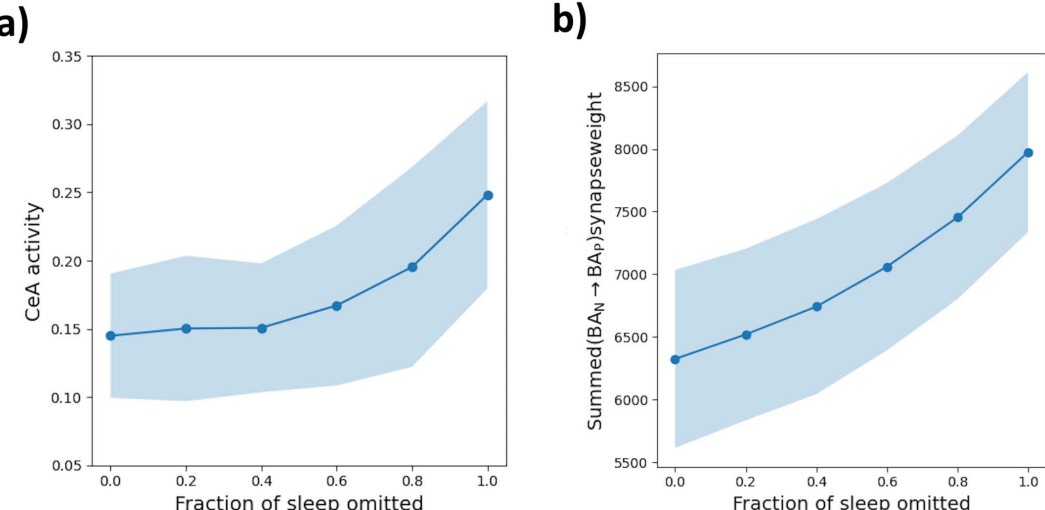

**Fig 8. Effects of chronic sleep deprivation Instances of the model were subjected to 7 days filled with exposure to various contexts, joined by US signals of random, generally low, strength.** At the end of the simulation, the model's fear response was assessed, after the brief exposure (3 time steps) to a moderate US signal (strength 0.6) in a novel environment. **a)** Instances of the model whose daily *Sleep* duration was systematically decreased (by a fraction of $20, 40, \cdots$, or 100%) acquire a greater fear response in the novel environment, on average. **b)** Sleep deprivation is further accompanied by net increases in the strength of synapses from context- onto fear-coding BA cells, measured at the end of the simulation protocol. While increases in fear acquisition only occurr after relatively drastic decreases in *Sleep* length, the average $BA_N \rightarrow BA_P$ synaptic density increases more linearly. In both plots, dots denote the mean, shaded regions one standard deviation across 30 runs of the simulation.

acquisition – relative to the baseline model and to shorter *Sleep* durations, though not relative to a complete lack of *Sleep*. This indicates that *Sleep*'s de-sensitizing effect on fear acquisition relies on intact synaptic pruning.

In summary, our model predicts that sleep disruptions – especially after aversive experiences – favour net increases in amygdalar synapse density, promoting fear expression and accelerating future acquisition. This suggestion highlights synaptic sleep homeostasis as a potential mechanism linking sleep disruptions to anxiety symptoms in disorders such as PTSD [94,95]. With regard to the proposal of avoiding sleep in the aftermath of trauma, our model indicates that this strategy may, irrespective of any potential effect on memory consolidation, pose the risk of aggravating the event's impact on the brain's fear circuitry.

**Stress-enhanced fear learning.** In the previous section, we demonstrated that depriving our model of *Sleep* disrupts synaptic homeostasis in the amygdala, allows weak synapses to accumulate and thereby favours a fear-sensitized state. Further, we noted that altering the model's homeostasis rule, making $BA_N \rightarrow BA_P$ synapses less likely to be pruned, interferes with *Sleep*'s moderating effect on later fear acquisition.

In reality, synaptic pruning is influenced by a wide range of environmental and internal factors [96,97]. Of particular interest is the finding that psychological stress can interfere with processes of synaptic weakening in hippocampus and amygdala [98,99], perhaps linked with aberrant synthesis of the stress hormone noradrenaline (NE) [100]. In the following, we incorporate neural effects of psychological stress into our model, enabling it to reproduce several aspects of the experimental paradigm known as Stress-Enhanced Fear Learning (SEFL) [22,23].

SEFL describes the phenomenon whereby exposure to severe stress in one context ('A') subsequently enhances fear acquisition in an unrelated context ('B') compared to non-stressed controls [22]. Notably, enhanced fear occurs specifically when stress precedes – but not when it follows – mild fear conditioning [22], indicating a primarily non-associative fear enhancement effect rather than associative fear generalisation [23]. Empirically, extreme acute or chronic stress reliably induces lasting increases in long-term potentiation within the amygdala, potentially priming neural circuits for heightened fear learning and generalized fear responses [99,101,102].

We modelled such stress-induced disruptions by temporarily increasing the *recruitability* of fear-promoting (P-) cells, thus enhancing their likelihood of forming lasting associations with $BA_N$ context cells. Additionally, we lowered the *extinction threshold* of the homeostasis rule acting on $BA_N \rightarrow BA_P$ synapses, similarly favouring synaptic strengthening over pruning during *Sleep*. These parameter changes were activated only by prolonged exposure to sufficiently intense aversive events (cf. S2 Appendix), conditions that did not occur in our previous simulations, making the present results complementary and independent. After the cessation of extreme stress, parameters gradually returned to their default values over the course of around five simulated days.

We tested this implementation using a protocol designed to capture key elements of the SEFL paradigm [22,23]. First, the model was exposed to extreme stress (high-intensity US) in context 'A', followed by a *Sleep* phase. On the next day, the model underwent mild fear conditioning (brief, moderate-intensity US) in a novel context 'B', again followed by *Sleep*. Finally, fear recall in context 'B' was measured. Consistent with experimental observations, this protocol resulted in enhanced fear expression in context 'B' compared to unstressed controls (Fig 9a). The SEFL effect did not occur when extreme stress occurred *one day after* moderate fear conditioning (Fig 9b) – matching empirical findings [22] – because, in this case, the parameter changes neither affected acquisition in 'B', nor replay of 'B' in the subsequent *Sleep* phase. Furthermore, the enhancement of fear acquisition was persistent, which we assessed by interposing 15 *Perception-Sleep* cycles involving mildly aversive experiences between conditioning in contexts 'A' and 'B' (Fig 9c). Previously stressed animals still acquired more fear in context 'B' because, although the described increases in P-cell recruitability had vanished by the time of conditioning, the associated net increases in synaptic strength between $BA_N$ and $BA_P$ persisted throughout the simulation (Fig 9d). This result aligns with the experimentally observed long-lasting impact of traumatic stressors in rats [16].

Thus, our model proposes a unified principle: impaired homeostatic pruning of amygdala synapses – whether due to sleep deprivation or severe psychological stress – may predispose amygdala circuits to maladaptive fear sensitization. Clinically, this hypothesis – if validated – would suggest that therapeutic or pharmaceutical interventions targeting either sleep quality or resilience to stress-induced homeostatic dysregulation could reduce vulnerability to pathological fear and anxiety disorders.

## Discussion

In this paper, we proposed a biologically inspired neural network model of contextual fear conditioning (CFC) that integrates sleep-dependent memory consolidation. We posit neural replay during sleep to be the driving mechanism behind the transfer of fear associations from hippocampal-amygdalar to amygdalo-cortical circuits. By assuming that cortical context representations overlap more strongly than hippocampal ones, our model naturally reproduces empirically observed increases in fear generalisation after CFC. In our simulations, the net synaptic strength between context- and fear-coding amygdala cells provided a heuristic for the model's 'fear sensitization' – an overall tendency towards acquiring and expressing fear. Drawing on the synaptic homeostasis hypothesis of sleep, we argue that impaired synaptic weakening in amygdalar fear circuits, as might plausibly occur due to sleep disruptions or stress, could contribute to such sensitization. Below, we discuss evidence, predictions and limitations of our account of memory consolidation, synaptic homeostasis and its impairment under stress.

### Sleep replay and circuits for fear memory consolidation

CFC experiments with rats have shown that forming a context-fear association involves the activity of memory-specific 'engrams' – cell populations later required for recall – in the hippocampus (HIP), medial prefrontal cortex (mPFC) and basolateral amygdala (BLA). Consistent with the standard model of systems consolidation [104,105], hippocampal engrams become functionally silent within two weeks [8,20]. In contrast, the mPFC engram remains silent while the memory is recent but matures in the weeks following its formation, eventually supporting remote memory recall [20,106–108].

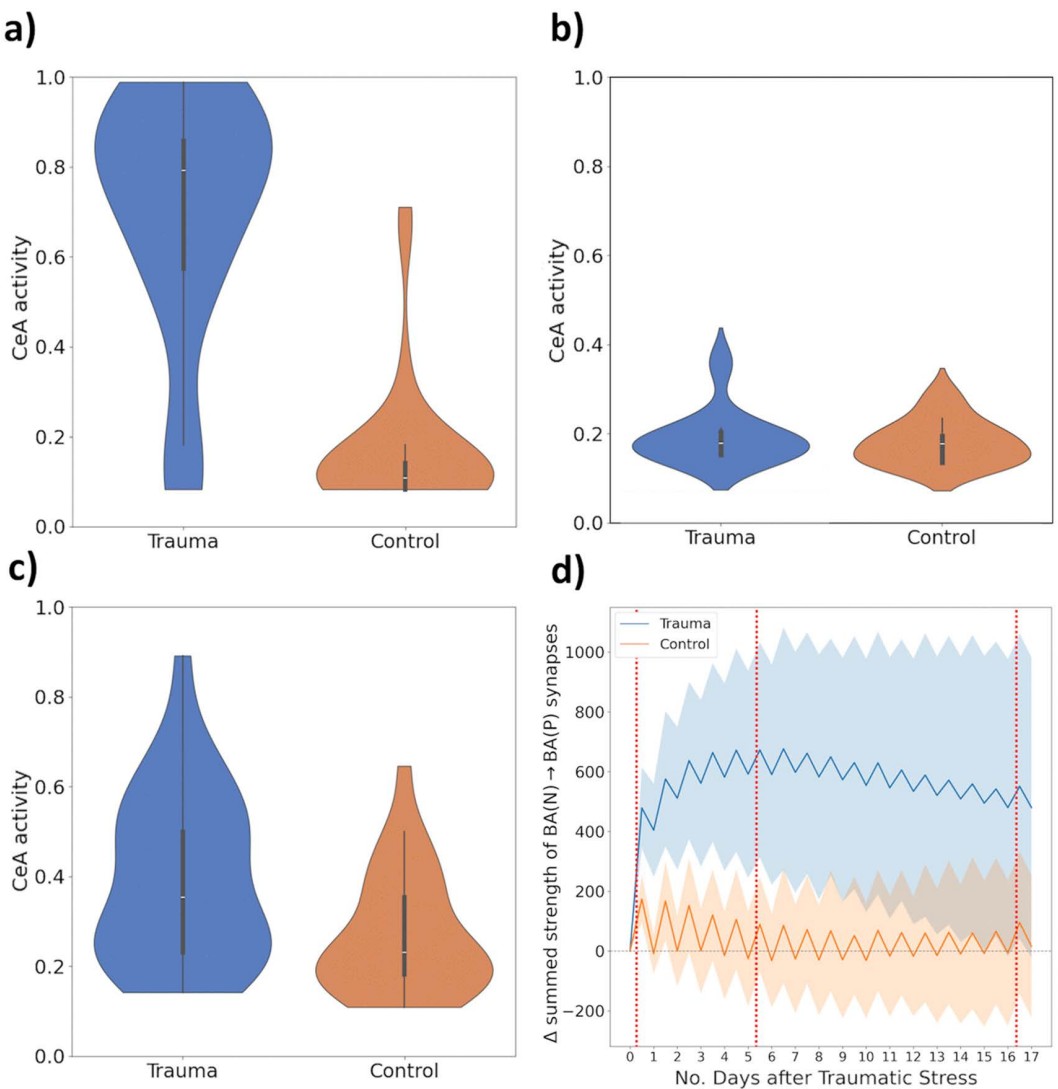

**Fig 9. Results on stress effects. a) Stress-Enhanced Fear Learning (SEFL).** Violin plots describe the magnitude of the model's fear response to context 'B', following the SEFL protocol as described in the main text, across 30 runs – with exposure to 'traumatic stress' on the one hand and without ('control') on the other. The trauma group acquires stronger fear responses to context 'B' (3 time steps, US strength 0.6), showing fear sensitization. The plotted density estimates were cut at the empirical maximum/minimum across the 30 runs. **b) SEFL does not occur if fear learning precedes stress.** Same as **a)**, but with the order of 'traumatic stressor' and 'moderate US' exposure reversed. Trauma exposure does not retrospectively enhance fear of context 'B'. **c) Stress enhances fear learning weeks later.** Same as **a)**, but with 'moderate fear acquisition' in context B carried out 15 *Perception-Sleep* cycles *after* traumatic stress. During the delay, the model meets various contexts (each paired with a US signal of generally low strength $\sim 0.9 \times$ Beta$(1, 2.5)$). Both groups show increased fear acquisition after the delay, due to fear generalisation; the effect is stronger the trauma group. **d) Stress raises the net strength of $BA_N \rightarrow BA_P$ synapses.** Blue and orange lines show the *total change* in the summed strength of $BA_N \rightarrow BA_P$ synapses of the 'trauma' and 'control' model instances, since the time of trauma exposure. Red vertical lines denote, from left to right, delivery of the traumatic stressor, the approximate time by which the 'trauma' model's parameters have recovered from stress, and delivery of the 'moderate US'. Values were recorded at the end of each *Perception* and *Sleep* phase; both curves oscillate since *Sleep* promotes a synaptic weakening [103]. On average, the $BA_N \rightarrow BA_P$ synaptic density of the trauma model shows a drastic increase on the day of the trauma, continues increasing over the following *Perception-Sleep* cycles, and remains far above that of the control model for the remainder of the simulation. Shaded regions denote $\pm 1$ SD across runs; variability results, e.g., from the random sampling of US signals over the course of the simulation.

This strengthening of cortical synapses relies on sustained hippocampal inputs [20,109], supported, e.g., by hippocampal sharp-wave ripple (SPW-R) events that primarily occur at rest [110–112]. Experimental manipulations indicate that mPFC-to-BLA projections become essential for memory expression approximately two weeks after CFC in rats and, notably, that the implicated BLA cells strongly overlap those previously targeted by HIP [20]. Consistent with this, recent findings have suggested a gradual strengthening of synapses between mPFC and BLA engram cells encoding a context-fear association, throughout the month following its formation [106]. Earlier studies had noted a strengthening of direct cortical-to-amygdalar synapses hours after learning [113,114].

Notably, the above time dynamics are specific to CFC. In contrast, the recall of fear associated with an elementary cue, such as an auditory tone, is independent from the hippocampus shortly after learning already, and engages a multisynaptic cortical-to-amygdalar pathway involving the paraventricular thalamus (PVT) [115,116]

Our implementation of *Sleep* replay in *HIP*, driving systems consolidation by reactivating newly formed long-term memory patterns in *CTX*, was adopted from Fiebig & Lansner (2014) [18]. The postulate that hippocampal replay during offline periods serves the role of 'instructing' the neocortex in this manner, transferring the responsibility of driving memory recall, is a longstanding tenet of the standard two-stage model of memory [117]. Since recalling a memory involves neural activity in a wide network of brain regions, it can be assumed that cortical-to-subcortical synapses must be strengthened as part of this memory reorganization [118]. This understanding, together with the evidence outlined above, informed our model of contextual fear memory consolidation. We assume that, in CFC, neuromodulation driven by prediction errors allows the amygdala to form a recurrently connected engram, which attains a valence by recruiting fear output neurons. This engram is initially tied to the hippocampus, and must be co-activated with neocortical engram cells to preserve the conditioned fear association. A test of this mechanism, within a CFC paradigm, would involve identifying the post-learning window during which co-activation of engram cells in neocortex and amygdala emerges. A next step may be to optogenetically disrupt plasticity in cortico-amygdalar pathways [119], or to selectively prevent reactivation of the amygdalar engram in this time window [7]. Our model predicts that this would reduce fear expression during recall tests at remote, but not recent, time points, without preventing recall of the conditioned context itself.

### Sleep replay and fear memory consolidation: Limitations

Although coordinated hippocampus-mPFC-amygdala activity after fear learning has been reported [9,120], temporally precise, across-region engram co-activations during sleep have not yet been demonstrated. The true dynamics of cross-regional replay are more complex than our model can account for, involving, e.g., nested or rapidly alternating ('flickering') hippocampal replay patterns during sleep [121]. Relaxing the assumption that hippocampal and amygdalar patterns are reactivated synchronously may naturally account for memory-linking effects, whereby neutral memories formed close in time to an aversive event, but in a different context, gradually acquire negative valence [122]. For instance, this may hold if hippocampal 'flickering' could cause a *BA* ensemble to co-activate with distinct *HIP* engrams in quick succession. Memory linking may also relate to an increased neuronal overlap between hippocampal memories formed close in time [52,123,124], which could be explained by a neuron-specific mechanism similar to our model's '$BA_{PII}$ cell recruitability'.

Clarifying the temporal organization of cross-regional replay and the synaptic mechanisms underlying systems consolidation remain key goals for future experiments. Regarding timescales, our replay implementation after Fiebig & Lansner (2014) assumes that several recently formed hippocampal memories are reactivated within a 'night' [18] – with each replay event lasting several simulation steps, but leaving the time step duration indeterminate. This is a functional abstraction; we do not model the oscillatory dynamics that may orchestrate these replay events (e.g., sharp-wave ripples [9]), nor the temporal arrangement of hundreds of these events over the course of Rapid Eye Movement (REM) [125] and slow-wave sleep [126].

Further, it should be noted that the 'standard theory of memory consolidation' – to which our model subscribes by positing that transient, hippocampal memory traces guide the strengthening of persistent, neocortical representations – has recently been scrutinized (cf. [127] for a detailed review). For instance, future models may reconsider our simplistic assumption that consolidation is unidirectionally driven by $HIP \rightarrow CTX$ and $HIP \rightarrow BA$ synapses. A $CTX \rightarrow HIP$ feedback mechanism could, e.g., be used to account for reconsolidation; a cognitive process relevant for exposure-based therapies [128]. In short, reconsolidation describes a process in which recalling a (cortical) long-term memory re-engages hippocampal circuits, temporarily rendering the memory modifiable before it is restabilised [129].

Similarly, there is a range of findings hinting at computational roles for projections from the amygdala to the hippocampus [130], which our model does not account for. For instance, the hippocampus does not only encode an animal's environmental-, but also interoceptive context [131] and internal-state cues can indeed serve as conditioned stimuli for fear conditioning [132]. $BA \rightarrow HIP$ feedback may further be hypothesized to contribute to the enhanced storage of emotional memories [133], e.g., by biasing hippocampal sleep replay towards memory traces associated with harm [9].

Moreover, replay in the biological brain is not exclusively hippocampal, but has been found to occur independently, e.g., in the dorsal striatum [134] and visual cortex [135]. In relation to this, we should note that, while cortical memory traces in our model decay at a much slower rate than hippocampal ones, they are not eternal (cf. Fig 4c). Hence, our model would predict that cortical traces, too, need to be occasionally replayed, though with less temporal urgency, to be permanently retained. Neural replay in memory systems beyond the hippocampus is an intriguing possibility [136]. It may protect synaptic engrams from degradation in conjunction with other, *biochemical* positive feedback loops for, e.g., sustaining protein activation or synthesis [63,137].

## Sleep homeostasis, stress and fear sensitization

Many existing neural network models of CFC rely on synapses connecting context-coding neurons, activated by environmental inputs, to valence-coding neurons driving defensive responses [10,11,19,41,138]. Although the assumption that context- and valence-coding neurons form distinct, clear-cut populations is likely an oversimplification, it provides a valuable conceptual framework for understanding how fear associations are stored.

To investigate aspects of fear memory *consolidation*, some simulations in the present paper extend over several weeks, during which our model perceives a range of different environments. Thus, the model had to be able to associate valence with a relatively large number of contexts in parallel, with little interference. To enable this, our model distinguishes itself from its predecessors by including larger populations of valence-coding cells, different subsets of which are recruited at different times [139].

A central tenet of the synaptic homeostasis hypothesis of sleep is that synaptic strengths accumulated during waking experiences undergo global down-regulation during subsequent sleep periods [54]. Evidence from cortical circuits suggests that this down-regulation preferentially prunes weak synapses while preserving stronger, functionally significant connections – potentially normalizing their strength rather than eliminating them [55,56].

Here, we have applied these principles to the synapses encoding context-fear associations in our model. The form of our model's fear circuitry, with the strengthening of valence-coding synapses based on an oscillating, postsynaptic *recruitability* factor, implied two computational roles that can be carried out by a synaptic homeostasis mechanism:

Normalizing strong synapses prevents excessive fear generalisation.

In our simulations, synapses linking context-coding cells to fear-coding neurons form with varying strengths influenced by existing synaptic weights, cell-intrinsic recruitability, and the intensity of aversive experiences (cf. Fig 7a). In particular, synapses that become part of two separate fear associations tend to grow stronger than required for the recall of either one. Within our framework, the strongest of these synapses promote maladaptive fear (over)generalisation by driving postsynaptic activity with little specificity. Sleep-dependent synaptic homeostasis may play a key role in this regard, keeping the strength of fear-coding synapses in check.

Regular pruning prevents build-up of synaptic strengths.

Our simulations further predict that regular pruning of weak, functionally insignificant synapses in amygdalar fear circuits – formed as a by-product of fear learning, e.g., due to spikes in *recruitability*, or caused by benign daily experiences – is critical to avoid cumulative increases in net synaptic strength. Without this pruning, fear-coding neurons may receive a gradually increasing amount of 'baseline excitation' in any unconditioned context. Consequently, new fear associations may form more rapidly and be expressed more strongly, even in the presence of merely moderate threat signals.

Sleep deprivation – whether acute or chronic – increases amygdala reactivity to negative emotional stimuli [58,140–142]. In humans, this effect is associated with emotional hyperreactivity and is commonly attributed to impaired top-down regulation by the mPFC. Consistent with this, sleep deprivation prior to fear conditioning experiments has been linked with increased fear expression during training and accelerated fear acquisition [90,143]. We hypothesize that sleep regulates future emotional reactivity, including fear expression, by decreasing synaptic strengths in excitatory input pathways onto amygdala cells responsive to aversive stimuli.

Both rodent and human studies further support the prediction that sleep omission in the immediate aftermath of fear conditioning impairs the behavioural discrimination between conditioned and safe stimuli (or contexts) [83,144–146]. In our model, as per Fig 7, the beneficial effect of sleep in this regard relies on a normalization of synapses implicated in recent fear learning events – and a pruning of redundant synapses – consolidating learned associations to support robust recall, while limiting maladaptive generalisation. It is hypothesized that low levels of certain neuromodulators (e.g., noradrenaline, serotonin and histamine) are crucial for synaptic weakening during sleep [54]. Our model thus yields the testable hypothesis that augmenting these neuromodulators overnight should reproduce effects similar to sleep deprivation, enhancing subsequent fear acquisition or reducing the specificity of learned associations.

Notably, psychological stress is known to enhance the release of noradrenaline and has been shown to promote long-term potentiation (LTP) relative to long-term depression (LTD) within the amygdala [99,147,148]. In line with the above, chronic stress in the days and weeks before fear conditioning experiments has indeed been linked with heightened excitability in amygdala circuits and enhanced fear acquisition [149,150]. Although acute stress caused by ongoing experiences similarly increases amygdala excitability, suggesting a direct influence of the noradrenergic system on emotional reactivity [151], our model further predicts that stress drives persistent plastic changes in amygdala fear circuits by disrupting overnight processes of synaptic weakening. Within our framework, synaptic net increases in pathways providing excitation to fear-coding neurons may manifest as a form of context-independent fear sensitization during future fear learning. They can be long-lasting as, once consolidated, synapses whose strength has accumulated under stress would not be homeostatically pruned once stress ceases.

In summary, our model suggests that psychological stress impairs fear regulation by disrupting a sleep-dependent synaptic weakening in amygdala fear circuits. If experimentally confirmed, this mechanism could explain how acute stress can persistently alter fear expression and learning, potentially clarifying one pathway by which stress exposure increases vulnerability to anxiety or trauma-related disorders in susceptible individuals.

## Conclusion and future directions

Here we have presented a neural network model for the formation and consolidation of associative context-fear memories. Our model suggests that the maturation of a cortical–amygdalar engram for the remote recall of contextual fear memories [20] relies on simultaneous co-activation of cortical and amygdalar engram cells, driven by hippocampal replay events during sleep. The model further predicts that the long-term evolution of an aversive memory depends on whether the involved amygdalar synapses are stabilised or pruned during offline memory consolidation. We hypothesize that a stress-induced failure of synaptic pruning in the aftermath of a fear learning event favours the accumulation of synaptic strengths in amygdala circuits and may thereby contribute to a chronic fear-sensitised state.

By incorporating sleep-dependent consolidation processes, our model expands on existing computational models of fear learning and provides clear hypotheses for experimental validation. Clarifying the putative links between psychological stress, disruptions in synaptic homeostasis, and subsequent emotional sensitization is a task with clinical relevance. For instance, heightened fear sensitization, excessive generalisation, and inflexible emotional learning – potentially resulting from saturation of synaptic plasticity [152] – are hallmark symptoms of PTSD [102,153,154]. These symptoms frequently co-occur with chronic sleep disruptions, particularly in REM sleep [86]. Relevant to this, the noradrenaline-producing locus coeruleus (LC) is known to be hyperactive in PTSD [155], potentially mediating hyperresponsiveness to threatening stimuli [156]. LC projections to the amygdala are amplified by psychological stress [157], and their absence during healthy REM sleep likely facilitates critical processes of synaptic weakening [103,158].

Given its broad scope, the current model implements several processes – including synaptic homeostasis, stress-induced plasticity changes, and neuronal allocation – at a relatively high level of abstraction. Future computational studies can leverage this qualitative framework to explore specific biological mechanisms more deeply, including neuromodulatory dynamics (e.g., noradrenergic modulation via the LC), cellular-level plasticity mechanisms, and molecular pathways underlying synaptic homeostasis. Addressing these detailed mechanisms is critical, as the neural underpinnings of fear memory processing and their dysfunction in neuropsychiatric disorders remain sparsely understood. Increasing biological specificity in computational models will facilitate the generation of experimentally testable predictions, enhance interpretation of empirical findings, and support the development of targeted interventions. Pursuing these directions is essential for bridging computational neuroscience with clinical advances in treating fear-related disorders.

## Supporting information

**S1 Appendix. Formal Model Description.**
(PDF)

**S2 Appendix. Full Update Cycle of the Model.**
(PDF)

**S3 Appendix. Stability Analysis.**
(PDF)

**S4 Appendix. Simulation Protocols.**
(PDF)

**S1 Fig. Memory formation and recall in HIP.** Any engram formed in the HIP module becomes associated with a pattern in the EC output layer, equal to the input pattern that led to its formation. Dotted arrows denote plastic synapses being strengthened. When the engram is retrieved by the same (or a sufficiently similar) cue, the original input pattern is activated in the output layer and observed to match the current input. In more detail: During *Perception*, the context-defining activity pattern of the EC input module ($EC_{in}$) is *copied* to $EC_{out}$, and synapses between HIP and $EC_{out}$ are rapidly strengthened. During *Recall*, the direct projection from input- to output layer is switched off, but the gain of the HIP-to-$EC_{out}$ connection is switched on. If a memory was recalled, $EC_{out}$ should therefore contain the pattern that was present when that memory was formed, while the activity of $EC_{in}$ depends on the current sensory input. An F1 score is computed between $EC_{in}$ and $EC_{out}$ pattern; if it exceeds a certain threshold, recalled and current context are said to match. If a mismatch is detected, inputs from HIP to $BA_N$ are switched off and the responsibility of retrieving a $BA_N$ representation for the current environment falls to CTX.
(TIF)

**S2 Fig. P-cell recruitability. (i.)** Rows of the heatmap denote individual P-cells, columns denote time steps of our simulation. Rows of the heatmap were sorted according to the cells' 'phase' (cf. below) after 12,000 time steps. The blurring of the white vertical lines at prior time points indicates that P-cells change the 'partners' they are highly recruitable *with* over time. **(ii.)** Evolution of the recruitability for an individual P-cell, corresponding to the first row of plot (i). Time windows of high recruitability last about 200 time steps and are interspersed by considerably longer phases of low recruitability. **(iii.)** The total sum of the recruitability values of all P-cells remains fairly stable over time.
(TIF)

**S3 Fig. Fear renewal trajectories in multi-context extinction ().** In the multi-context *ABC* renewal paradigm from the main results (Fig 6b), we varied the overlap between *A* and the extinction contexts $B_i$ ($A \leftrightarrow B_i$=0.50, 0.59, 0.68, 0.77, 0.86, 0.95; $A \leftrightarrow C$=0.80). Each panel shows the average CeA output across acquisition (context *A*, blue), extinction ($B_1 \ldots B_4$, red), and renewal (*C*, green) phases for different overlaps. Renewal in *C* is weakest at intermediate similarity (=0.77, 0.86). Shaded areas denote ±SEM across runs ($n = 10$).
(TIF)

**S1 Text. An intermediate similarity between acquisition and extinction contexts minimizes fear renewal in a didactic toy model.**
(PDF)

**S2 Text. Effects of chronic sleep deprivation for different model variants.**
(PDF)

## Author contributions

**Conceptualization:** Lars Werne, Angus Chadwick, Peggy Seriès.

**Formal analysis:** Lars Werne.

**Methodology:** Lars Werne, Angus Chadwick, Peggy Seriès.

**Software:** Lars Werne.

**Supervision:** Angus Chadwick, Peggy Seriès.

**Validation:** Lars Werne.

**Visualization:** Lars Werne.

**Writing – original draft:** Lars Werne.

**Writing – review & editing:** Lars Werne, Angus Chadwick, Peggy Seriès.

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
