## [Decision Letter · Decision Letter 0]

19 Aug 2025

Learning, sleep replay and consolidation of contextual fear memories: A neural network model

PLOS Computational Biology

Dear Dr. Seriès,

Thank you for submitting your manuscript to PLOS Computational Biology. After careful consideration, we feel that it has merit but does not fully meet PLOS Computational Biology's publication criteria as it currently stands. Therefore, we invite you to submit a revised version of the manuscript that addresses the points raised during the review process.

Please submit your revised manuscript within 60 days Oct 19 2025 11:59PM. If you will need more time than this to complete your revisions, please reply to this message or contact the journal office at ploscompbiol@plos.org. Please include the following items when submitting your revised manuscript:

We look forward to receiving your revised manuscript.

Kind regards,

Daniel Bush

Academic Editor

PLOS Computational Biology

Marieke van Vugt

Section Editor

PLOS Computational Biology

**Additional Editor Comments:**

The authors should clarify how robust their results are to different parameter settings, and the relationship between their model and empirical data, as described in more detail below.

**Journal Requirements:**

3) Please upload a copy of Figures ??a, ??d. Fig ??b, and Fig ??c which you refer to in your text on pages 46, and 47. Or, if the figure is no longer to be included as part of the submission please remove all reference to it within the text.

4) We notice that your supplementary Figures, and Tables are included in the manuscript file. Please remove them and upload them with the file type 'Supporting Information'. Please ensure that each Supporting Information file has a legend listed in the manuscript after the references list.

Potential Copyright Issues:

i) Figures 1, 3, and 5. Please confirm whether you drew the images / clip-art within the figure panels by hand. If you did not draw the images, please provide (a) a link to the source of the images or icons and their license / terms of use; or (b) written permission from the copyright holder to publish the images or icons under our CC BY 4.0 license. Alternatively, you may replace the images with open source alternatives. See these open source resources you may use to replace images / clip-art:

6) In the online submission form, you indicated that your data will be submitted to a repository upon acceptance. We strongly recommend all authors deposit their data before acceptance, as the process can be lengthy and hold up publication timelines. Please note that, though access restrictions are acceptable now, your entire minimal dataset will need to be made freely accessible if your manuscript is accepted for publication. This policy applies to all data except where public deposition would breach compliance with the protocol approved by your research ethics board. If you are unable to adhere to our open data policy, please kindly revise your statement to explain your reasoning and we will seek the editor's input on an exemption.

7) Please amend your detailed Financial Disclosure statement. This is published with the article. It must therefore be completed in full sentences and contain the exact wording you wish to be published.

2) If any authors received a salary from any of your funders, please state which authors and which funders..

8) Please send a completed 'Competing Interests' statement, including any COIs declared by your co-authors. If you have no competing interests to declare, please state "The authors have declared that no competing interests exist". Otherwise please declare all competing interests beginning with the statement "I have read the journal's policy and the authors of this manuscript have the following competing interests:"

**Reviewers' comments:**

Reviewer's Responses to Questions

**Comments to the Authors:**

Reviewer #1: The paper presents a biologically grounded neural network model of contextual fear memory formation and consolidation, organized into modules that mirror key brain regions. A sensory cortical module encodes environmental contexts and sends this information in parallel to both a hippocampal module, which encodes contexts rapidly via entorhinal inputs, and a neocortical module that gradually builds long term, generalized context representations through consolidation. A basal amygdala module encodes associations reflecting each context’s valence. Offline replay allows these representations to be strengthened in the neocortex. Replay supports both renewal and time dependent increases in generalization, while synaptic homeostasis trims weak associations to prevent overgeneralization. These dynamics also enable the model to account for stress enhanced fear learning and the maladaptive effects of sleep deprivation.

The manuscript is clear and well-organized, outlining each model component and its empirical rationale in a way that allows readers to understand how replay and synaptic downscaling contribute to the observed effects. The model offers a thorough, operationalized implementation of the hypothesis that fear generalization reflects hippocampal-to-neocortical transfer, providing a strong foundation for future experimental work. However, the conceptual novelty of the framework may be somewhat limited, as systems consolidation has long been proposed as a mechanism underlying generalization. The following comments raise several issues regarding the rationale behind specific architectural choices, the consistency of mechanistic assumptions across simulations, and the model’s alignment with related empirical findings on contextual fear memory consolidation.

Major comments:

1. There appear to be discrepancies between the model’s architecture and established anatomical pathways in the brain. In the model, the sensory cortex feeds separately into the hippocampus and neocortex. However, in the brain, the sensory cortex is part of the neocortex, and hippocampus inputs are processed through neocortical hierarchies that interact with the hippocampus via a bidirectional loop. The inclusion of ECout as an isolated output in the model also departs from the brain, as the entorhinal cortex receives hippocampal outputs and projects them back to the neocortex. The authors should clarify or revise these architectural choices to better reflect the relevant circuitry.

2. The benefits of consolidation for memory retention appear limited in the current model. For example, Figure 4c shows that consolidated cortical memories decay linearly over time and would eventually be lost, suggesting that consolidation provides only a short-term advantage. This seems like an important limitation. The authors should consider discussing or implementing mechanisms that help stabilize cortical representations over longer timescales.

3. Greater consistency in mechanistic assumptions across simulations would strengthen the model. Currently, some implementation details appear only in specific modules or simulations, making it unclear which assumptions apply universally. For instance, it is not clear whether the suppression of fear-inhibiting BA_I cells is learned or hard-coded, whether the higher activation threshold for BA_I versus BA_P cells holds across simulations, or whether the boost in cell recruitability applies only to the amygdala or more broadly. Clarifying which mechanisms are implemented universally and which are specific to particular simulations or model components, and specifying which results rely on these assumptions being simulation- or component-specific, would help readers better interpret the results.

4. The model’s ability to align with related empirical findings on fear memory consolidation remains unclear. For instance, studies have shown that hippocampal replay selectively prioritizes fear-related experiences (Wu et al., 2017), and that strong aversive events can retroactively strengthen neutral memories formed days earlier (Zaki et al., 2025). In the current model, replay and hippocampal-to-amygdala connections do not vary with valence, and valence does not modulate replay. As a result, it is not obvious whether the model could account for such findings. It would be valuable to discuss or explore potential mechanisms that might allow the model to capture these effects.

Minor comments:

1. The claim “Ours is the first to incorporate a neural mechanism enabling” fear memory consolidation could be softened, given that models such as Mattar & Daw (2018) also simulated consolidation of fear memory. A more accurate phrasing might be that this is the first model to implement systems consolidation of fear memory.

2. In Figure 1 (page 5), the legend refers to “Nodes labelled with an S-shape”. I am not sure which nodes are marked that way.

3. On page 9, “In the following, we …” seems incomplete.

4. Also on page 9, please add a space in “inRecall mode” so that it reads “in Recall mode.”

5. In Figure 5c (page 13), what does the y-axis label “CeM” refer to?

6. When discussing insomnia and sleep disruptions (page 14), the motivation would be clearer if these pathologies were tied more directly to fear acquisition and expression.

7. The Kitamura et al. study is described in detail in the Discussion, where readers might expect a higher-level summary. The authors might consider introducing it earlier to help motivate the model’s architecture, or alternatively, making the description in the Discussion more concise.

Reviewer #2: The authors extended a previous computational model proposed by Fiebig and Lansner to address the underlying neural mechanisms of learning and consolidation of contextual fear memories. This model consists of three networks: the hippocampus, the neocortex, and the amygdala. Their simulations provide a mechanistic account of the observed transfer of context–fear memories from hippocampal–amygdalar to amygdalo–cortical circuits. The model also reproduces empirically observed increases in fear generalisation following contextual fear conditioning. In addition, they found that (in their model) impaired synaptic weakening in amygdalar fear circuits—as might plausibly occur due to sleep disruptions or psychological stress—could mechanistically contribute to fear sensitisation. Overall, this is a valuable model, and experimentally testing some of its hypotheses could yield clinically relevant insights.

Major comments:

1, This work focuses mainly on how fear memory acquisition can be enhanced and generalised. This is valuable, as it helps to elucidate potential neural mechanisms underlying these phenomena through computational modelling. However, what is perhaps more interesting—and more relevant for translational neuroscience—is how established fear memories can be weakened, particularly traumatic memories. I would like to see some simulations targeting this point included in the paper. There is a short section on fear extinction, but it is too brief and lacks sufficient detail on how sleep, de-stressing, or other potential factors may lead to fear memory weakening/extinction.

For example:

• Line 334: “…enhancing synaptic sleep homeostasis in the amygdala could be a key therapeutic strategy in PTSD and related disorders.” I would expect the authors to demonstrate this directly in their model.

• Figure 7: the authors show that chronic sleep deprivation leads to accelerated fear acquisition. I would also expect to see conditions that decelerate fear acquisition.

2, I have concerns about the modelling of hippocampal replay. From Ji & Wilson, we know that replay coordinates sequential activity between the hippocampus and visual cortex, but this occurs during specific windows of sleep. In the current context, “replay” is used to establish communication between hippocampus, cortex, and amygdala. This does not align precisely with the timescale of replay, which is usually tens to hundreds of milliseconds. For example, it is unclear what the unit of the time step in Fig. 4b is. In the text you mention “over the course of a sleep phase”, but it is not clear what duration you mean by a sleep phase. Is it an entire NREM episode? At other times the timescale is given as “days” (e.g., Fig. 5d and Fig. 8d). The authors need to make the timescale consistent, or, if that is not possible, provide clearer descriptions.

3, I ran the authors’ provided code and was able to reproduce the figures in the main text. This is a relatively complex model with many parameters, and I give credit to the authors for making this model work as intended. However, I am somewhat concerned about the robustness of the reported results. For example, if a parameter setting were altered, would the results remain qualitatively similar? Or is the current configuration “cherry-picked”? A stability analysis of the model would strengthen confidence in the results.

4, Although the model explains previous empirical findings, makes testable predictions, and links these results to potential neural mechanisms, there remains a significant gap between the computational assumptions/mechanisms and the actual neural mechanisms underlying these phenomena. For example:

• Line 314: “Our model can explain these findings due to our assumption that sleep after fear learning prunes weak synapses onto P-cells.”

• Line 331: “Targeting sleep disruptions … may directly influence the synaptic mechanisms underlying chronic anxiety.”

• Line 383: “…implies that therapeutic interventions targeting either sleep quality or resilience to stress-induced homeostatic dysregulation could reduce vulnerability to pathological fear and anxiety disorders.”

These claims are somewhat too strong, given that they derive from a computational model rather than direct empirical evidence. The language could be moderated to avoid potentially misleading over-interpretation. I understand that some of these issues are difficult to address fully in a computational modelling study. Nonetheless, I find the model valuable.

Minor comments

1. Line 22: “…also exhibits activity coordinated with hippocampal replay during sleep…” — needs citations.

2. Is there empirical evidence supporting the choice to model CTX as a BPCNN and HPC as a WTA network? How do these differ from a Hopfield network, and why was a Hopfield network not used instead?

3. In the author summary, the authors ask: “Why do some fear memories fade while others persist or even grow stronger over time?” After reading the paper, I did not find a clear answer. Can the model explain this? (This relates to my major comment 1.)

4. Line 76: the authors state that representations in the hippocampus are short-lived and overwritten by new learning. In Hopfield-like models, this would be termed catastrophic forgetting. Why is this not the case for the neocortex in the proposed model?

5. Line 79–80: replay of sequential patterns is implemented via inhibitory synapses operating at a fast timescale. Previous works (e.g., Azizi et al., 2013; Ji et al., 2024) used adaptation to generate replay-like dynamics. What is the difference between these approaches, given that the present model does include adaptation (Eq. 7–10)?

6. Line 239: “…fear extinction depends primarily on silencing, rather than erasure…” The model shows that silencing can work, but why not erasure? This was not tested.

7. Fig. 4a: why does the hippocampus rapidly form context engrams? Is this due to fast synapses?

8. Fig. 4d: why is the x-axis labelled “trials” rather than “steps”?

9. Line 300: what does “unchecked” mean here?

**Have the authors made all data and (if applicable) computational code underlying the findings in their manuscript fully available?**

Reviewer #1: **No:** The authors include the following note: "All code written in support of this publication will be publicly available on Zenodo."

Reviewer #2: Yes

PLOS authors have the option to publish the peer review history of their article (what does this mean? ). If published, this will include your full peer review and any attached files.

**Do you want your identity to be public for this peer review?** For information about this choice, including consent withdrawal, please see our Privacy Policy .

Reviewer #1: No

Reviewer #2: **Yes:** Zilong Ji

**Figure resubmission:**

**Reproducibility:**



---

## [Decision Letter · Decision Letter 1]

7 Jan 2026

PCOMPBIOL-D-25-01216R1

Learning, sleep replay and consolidation of contextual fear memories: A neural network model

PLOS Computational Biology

Dear Dr. Seriès,

Thank you for submitting your manuscript to PLOS Computational Biology. After careful consideration, we feel that it has merit but does not fully meet PLOS Computational Biology's publication criteria as it currently stands. Therefore, we invite you to submit a revised version of the manuscript that addresses the points raised during the review process.

We look forward to receiving your revised manuscript.

Kind regards,

Daniel Bush

Section Editor

PLOS Computational Biology

Marieke van Vugt

Section Editor

PLOS Computational Biology

**Additional Editor Comments:**

Both reviewers are happy to accept the manuscript for publication, but Reviewer 2 has requested a few very minor final edits. I think it is important to make these changes, but I will check those myself - the manuscript will not go back out for review.

**Journal Requirements:**

**Reviewers' comments:**

Reviewer's Responses to Questions

**Comments to the Authors:**

Reviewer #1: Overall, the authors’ responses are clear and thorough, and the revisions address most of the concerns. The architectural abstractions are now better motivated, the limitations of cortical memory decay are acknowledged, and parameter consistency across simulations is clarified. The discussion of valence-biased replay and memory linking is appropriate, even if these remain future extensions. All minor comments were handled cleanly. Together with the revisions made in response to Reviewer #2, the manuscript is substantially improved, and the authors’ responses are generally satisfactory.

Reviewer #2: Thank you for the detailed responses to my review. I am satisfied with most of the authors’ replies, particularly the inclusion of parameter sweeps, which strengthen the complex network model. Overall, I consider this work to be valuable and have recommended it for publication.

However, some of the network manipulations remain relatively coarse and lack detailed neurobiological grounding. As a result, there is still a gap between the modelled phenomena and their biological interpretation, and alternative computational mechanisms might plausibly lead to similar results.

I have a few additional suggestions:

1. For figures reporting results averaged over multiple simulation runs, measures of variability (e.g. s.e.m. or s.d.) should be included in the plots, for example in Fig. 4.

2. Line 31: “in PTSD — often emerge gradually, rather than immediately, after intense fearful or traumatic experiences.” Appropriate citations are missing here.

3. Line 50: citations 22–24 do not seem necessary in this context.

4. Line 90: please consider adding a citation to the Hopfield (2010, PNAS) paper on mental exploration.

**Have the authors made all data and (if applicable) computational code underlying the findings in their manuscript fully available?**

Reviewer #1: None

Reviewer #2: Yes

PLOS authors have the option to publish the peer review history of their article (what does this mean? ). If published, this will include your full peer review and any attached files.

**Do you want your identity to be public for this peer review?** For information about this choice, including consent withdrawal, please see our Privacy Policy .

Reviewer #1: No

Reviewer #2: No

**Figure resubmission:**
---

## [Editor Report · Decision Letter 2]

20 Feb 2026

Dear Prof Seriès,

We are pleased to inform you that your manuscript 'Learning, sleep replay and consolidation of contextual fear memories: A neural network model' has been provisionally accepted for publication in PLOS Computational Biology.

Best regards,

Daniel Bush

Section Editor

PLOS Computational Biology

Marieke van Vugt

Section Editor

PLOS Computational Biology
